# Language Model Behavioral Phases are Consistent Across Architecture, Training Data, and Scale

**James A. Michaelov[1,2,3], Roger P. Levy[1], Benjamin K. Bergen[3]**
[1] Department of Brain and Cognitive Sciences, MIT    [2] MIT Libraries CREOS
[3] Deparmtent of Cognitive Science, UCSD
{jamic,rplevy}@mit.edu,  bkbergen@ucsd.edu

## Abstract

We show that across architecture (Transformer vs. Mamba vs. RWKV), training dataset (OpenWebText vs. The Pile), and scale (14 million parameters to 12 billion parameters), autoregressive language models exhibit highly consistent patterns of change in their behavior over the course of pretraining. Based on our analysis of over 1,400 language model checkpoints on over 110,000 tokens of English, we find that up to 98% of the variance in language model behavior at the word level can be explained by three simple heuristics: the unigram probability (frequency) of a given word, the $n$-gram probability of the word, and the semantic similarity between the word and its context. Furthermore, we see consistent behavioral phases in all language models, with their predicted probabilities for words overfitting to those words' $n$-gram probabilities for increasing $n$ over the course of training. Taken together, these results suggest that learning in neural language models may follow a similar trajectory irrespective of model details.

## 1  Introduction

Language models are complex systems that ostensibly exhibit *emergent behaviors* (Anderson, 1972; Nicolis and Prigogine, 1977; Laughlin and Pines, 2000; O'Connor, 2021; Wei et al., 2022a). Trained only to predict the next word in a sequence given a context, language models learn to generate grammatical sentences, make predictions in line with real-world knowledge, and—given the right prompt—answer questions and exhibit reasoning-like behavior, even without finetuning (Linzen et al., 2016; Wei et al., 2022b,a; Biderman et al., 2023b; Hu and Frank, 2024; OLMo Team et al., 2025). How do they get there?

While language model learning involves gradual change by some metrics (Schaeffer et al., 2023), researchers have identified sudden shifts in model behavior (Du et al., 2024), as well as precipitous changes in model subnetworks that can drastically alter behavior (Olsson et al., 2022; Chen et al., 2024a). However, these analyses generally focus on specific, targeted behaviors or sub-networks. We focus more broadly on whether it is possible to characterize the overall behavior of models and how this changes over the course of training. We draw on two main lines of research. The first demonstrates that language model predictions are often sensitive to superficial properties of their input and output, such as training data, frequency of tokens, token co-occurrences, grammatical structures, tasks, and specific facts about the world; the baseline probability of input and output sequences and subsequences; and the degree of semantic similarity between an output word and its context (Forbes et al., 2019; Wei et al., 2021; Michaelov and Bergen, 2022; McCoy et al., 2024; Chang and Bergen, 2024). The second is previous work suggesting that language models overfit to $n$-grams of increasing $n$ over the course of training (Karpathy et al., 2016; Chang et al., 2024). These two lines converge to suggest that much of language model behavior can be characterized by relatively simple heuristics.

39th Conference on Neural Information Processing Systems (NeurIPS 2025).

We focus on three—frequency, $n$-gram probability, and semantic similarity. We ask to what extent they explain language model behavior at any point in time over the course of training, and how this changes. Our key contributions are the following:

1. We train and release the Parallel architecture (Parc) language models. These are made up of 1,314 language model checkpoints: 6 seeds each of models of the Pythia (Biderman et al., 2023b), Mamba-1 (Gu and Dao, 2024), and RWKV-4 (Peng et al., 2023) architectures, with each seed trained on the same OpenWebText (Gokaslan and Cohen, 2019) data, and each with 73 checkpoints taken over the course of training. As far as we are aware, these are the first available checkpointed Mamba and RWKV models.

2. We construct and release the Natural Words in Context (NaWoCo) dataset, an evaluation set of over 150,000 words in natural sentence contexts extracted from the FineWeb corpus (Penedo et al., 2024).

3. We show that three simple heuristics—frequency (i.e., unigram probability), $n$-gram probability, and semantic similarity to the context—explain up to 98% of the variance in the probabilities assigned to words in this (decontaminated) dataset by our models, as well as the Pythia (Biderman et al., 2023b; van der Wal et al., 2024) and Open-GPT2 (Karamcheti et al., 2021) language models.

4. We demonstrate that all the models tested (no matter the size, architecture, or training data) overfit to $n$-grams of increasing $n$ over the course of pretraining, and show a correlation with semantic similarity after the first 10–100 steps.

## 2 Related Work

### 2.1 $n$-gram-like prediction in language models

Neural language models learn representations that enable them to predict the next word in the sequence based on generalizations rather than simply storing counts of sequences like $n$-gram models (Markov, 1913; Shannon, 1948; Jelinek et al., 1975; Baker, 1975). However, there is evidence that both RNNs and transformers do still calculate the $n$-gram probabilities of words. Voita et al. (2024), for example, identify neurons in the OPT models (Zhang et al., 2022) that appear to be sensitive to a restricted number of specific $n$-gram—that is, they appear to be 'dedicated' (Voita et al., 2024) to these $n$-grams—which are more numerous in larger models. In addition, in contemporaneous work, Chang and Bergen (2025) identify bigram circuits in the Pythia models (Biderman et al., 2023b).

Behaviorally, the probabilities calculated by neural language models—both recurrent neural networks and transformers—have been found to be highly correlated with $n$-gram probabilities, especially early in training (Karpathy et al., 2016; Sun and Lu, 2022; Chang and Bergen, 2022; Choshen et al., 2022; Bietti et al., 2023; Voita et al., 2024; Akyürek et al., 2024; Nguyen, 2024; Chang et al., 2024; Belrose et al., 2024; Wang et al., 2025). A recent study by Chang et al. (2024), for example, found that over the course of pretraining, the probabilities calculated by 5 random seeds of a 117M-parameter GPT-2 model become differentially correlated with $n$-gram probabilities (calculated on the same training data) of increasing $n$ over the course of training—that is, they initially become more correlated with unigram probability (i.e. frequency) $n$-grams of $n > 1$, then become more strongly correlated with bigram probability than other $n$-grams, and so on until at least $n = 5$. In a related line of research, Nguyen (2024) found that for a given token prediction in a test corpus, it is possible to construct a rule based on a combination of $n$-grams in the training corpus of a language model that can make the same top prediction as the model with an accuracy of up 68–79%, depending on dataset and the procedure used to calculate the optimal rule for each token.

### 2.2 Similarity-based prediction in language models

The extent to which a word is similar to words in its context has also been shown to have an impact on the predictions of language models (Kassner and Schütze, 2020; Misra et al., 2020; Michaelov et al., 2021, 2024). One example of this is the finding that lexical semantic priming occurs in language models—Misra et al. (2020) find that prepending the word 'airplane' to 'I want to become a [MASK].' to create 'airplane. I want to become a [MASK].' increases the probability BERT assigns to 'pilot' being the masked token relative to both the original sentence and the sentence with a different

prepended word such as 'table'. In a less targeted form of analysis, Michaelov et al. (2021) find that GPT-2 surprisal (negative log-probability) has a Pearson correlation of $r = -0.48$ with the cosine similarity between the language model's static embedding of the same word and the mean of the embeddings of the words in the context; while Michaelov et al. (2024) find that GPT-3 (Brown et al., 2020) surprisal is correlated with the same similarity metric calculated using GloVe (Pennington et al., 2014) ($r = -0.46$) and fastText (Grave et al., 2018) ($r = -0.61$) word embeddings. Thus, while there is thus far no direct evidence that contextual semantic similarity plays an explicit causal role in language model predictions, there is at least strong evidence that it correlates with them.

# 3  Experiment 1: Correlations Between Language Model Log-Probabilities and Heuristics

In our first experiment, we investigate the extent to which language model probability correlates with $n$-gram probability and contextual semantic similarity over the course of pretraining. Going beyond previous work (e.g., Chang and Bergen, 2022; Choshen et al., 2022; Chang and Bergen, 2024; Nguyen, 2024; Belrose et al., 2024), we characterize exactly to what extent each $n$-gram log-probability for $n \in \{1, 2, 3, 4, 5\}$ is correlated with the log-probability assigned to a given word by language models of different architectures over the course of training. We also carry out the same analysis for semantic similarity as calculated using fastText word embeddings (Bojanowski et al., 2017; Grave et al., 2018). We carry out our analyses on a range of language models, which vary in architecture, size, and training dataset. All code, data, analyses, and models are provided in the following repository: `https://github.com/jmichaelov/lm-behavioral-phases`.

## 3.1  Method

### 3.1.1  Language Models

We carry out our analyses on two sets of pretrained language models with checkpoints taken over the course of their training. The first set were the **Pythia** models (14M–12B parameters; Biderman et al., 2023b) including the additional PolyPythia seeds (9 of each of the 14M–410M parameter models; van der Wal et al., 2024), all of which were trained on The Pile (Gao et al., 2020). The second were the GPT-2 models (5 seeds each with 117M and 345M parameters, of which we used 4; see Appendix A) trained as part of the Mistral project (Karamcheti et al., 2021) on the OpenWebText (Gokaslan and Cohen, 2019) corpus, henceforth the **Open-GPT2** models.

We also trained an additional 18 language models. Following the approach taken for the Open-GPT2 models (Karamcheti et al., 2021), we trained all models on 1024-token sequences of OpenWebText with a batch size of 512; though we train each for only 4,000 steps. In order to expand our analyses beyond transformers, we use the same 6 random seeds to train models with three architectures in parallel (in the sense that they encounter the same sequences at the same steps), using the same tokenizer for all models. The three architectures were the Pythia (Biderman et al., 2023b) transformer architecture, the Mamba-1 state-space model architecture (Gu and Dao, 2024), and the RWKV-4 architecture, which is a modern recurrent neural network architecture with parallelizable training (Peng et al., 2023). We henceforth refer to these as the **Parc-Pythia** (160M), **Parc-Mamba** (130M), and **Parc-RWKV** (169M) models, respectively.

Considering all models, seeds, and checkpoints, our analysis encompasses a total of 1,418 model instances. We provide the full code for training and running the models, and further details of all models used in Appendix A.

### 3.1.2  $n$-gram Log-Probability

To investigate the extent to which each language models' predictions match given $n$-grams, we calculate the $n$-gram probability of the same words in context for $n \in \{1, 2, 3, 4, 5\}$. We calculate $n$-grams using the same data as the model itself was trained on—that is, for the Pythia models, we calculate $n$-grams in The Pile, and for the OpenWebText models (Open-GPT2, Parc-Pythia, Parc-RWKV, and Parc-Mamba), we calculate $n$-grams in OpenWebText. Because the pretrained models have different tokenizers, we calculate all $n$-grams at the word (rather than token) level.

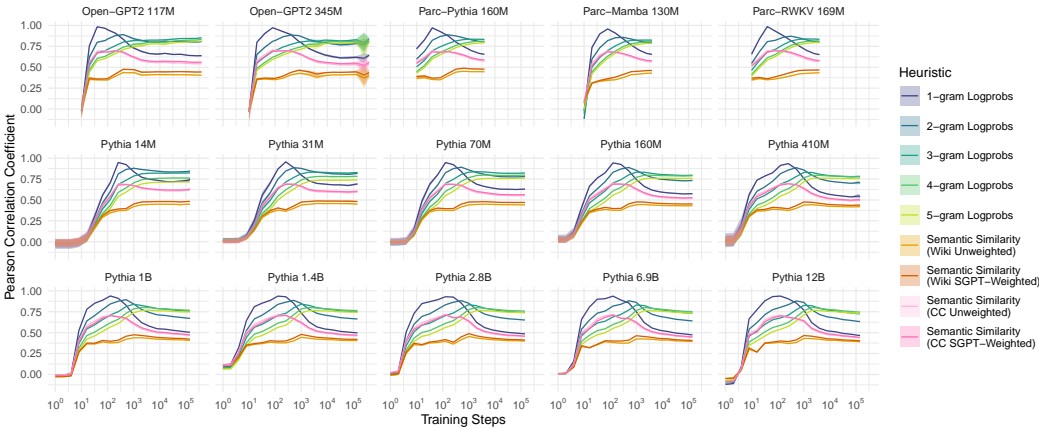

Figure 1: Pearson correlation coefficient $r$ between language model log-probability and heuristic metrics ($n$-gram log-probability and word embedding cosine similarity). We show the mean values for all models across seeds and their 95% confidence intervals.

To calculate $n$-gram probabilities, we use infini-gram (Liu et al., 2024), which allows a user to retrieve the total counts $c$ of a given sequence in a corpus. For The Pile, we use the infini-gram API to access word counts; and for OpenWebText, we build our own infini-gram index locally. We estimate $n$-gram probability based on these counts using *Stupid Backoff* as described by Brants et al. (2007). We provide the full code for building the index and calculating probabilities, and further details of our implementation in Appendix B.

### 3.1.3 Contextual Similarity

To calculate contextual similarity, we follow previous work investigating the relationship between word embedding similarity and language model log-probability (Michaelov et al., 2024). Specifically, we use fastText (Bojanowski et al., 2017; Grave et al., 2018) to calculate the word embeddings of each critical word and the words in their context, and take contextual similarity to be the cosine similarity between the embedding of the critical word and the mean of the embeddings of all words in the context. We calculate this for both the fastText embeddings trained on Wikipedia (Bojanowski et al., 2017) and those trained on Common Crawl (Grave et al., 2018). Additionally, for each of these, when calculating the context embedding, we calculate both the uniformly-weighted mean and a weighted mean based on the SGPT approach (Muennighoff, 2022). We provide further details of our implementation in Appendix C.

### 3.1.4 Evaluation Dataset

We carry out our analysis on a set of words in sentence contexts, which were all sampled from the FineWeb corpus (Penedo et al., 2024). Specifically, we constructed a set of words that were single tokens for all models, and that occurred as the fifth or later word in (unique) sentences that began with a capitalized word, had no other capitalized words, were assigned a probability of being toxic of 0.1 or less (using the model released by Logacheva et al., 2022), and, to avoid possible contamination, are not in the training data of any of the language models tested (based on infini-gram counts as described in Section 3.1.2). Our final dataset of Natural Words in Context (NaWoCo) is made up of a training set of 77,999 items, a validation set of 39,474 items, and a test set of 40,980 items. Further details of dataset construction are provided in Appendix D.

### 3.1.5 Analysis

To investigate the relationship between the heuristics ($n$-gram log-probability and semantic similarity) and the language models, we use each model at each training checkpoint to calculate the log-probability of every word in our training set. We then calculate the correlation between the log-probabilities of every transformer at every time step and each of the heuristics.

## 3.2 Results and Discussion

We show the Pearson correlation between each heuristic and language model log-probability in Figure 1 (for Spearman correlation, see Appendix E). As can be seen, one pattern is remarkably consistent across models—over the course of training, we see peaks in the correlation between transformer log-probability and $n$-gram log-probability for increasing values of $n$. As in Chang et al. (2024), we see that language models become more sensitive to higher-order $n$-grams as they continue to train. Going beyond past work, we see that this observation applies not only to transformers, but to autoregressive language models of non-transformer architectures as well. We also see that the pattern observed by Chang et al. (2024) can be is explained in our data by both an increase in the correlation with the higher-order $n$-grams over the course of training and a simultaneous decrease in the correlation with the lower-order $n$-grams—that is, the predictions move toward the former and farther away from the latter. In addition, as can also be seen in Figure 1, larger models also see a greater decrease in the correlation to smaller $n$-grams, suggesting that their probability distributions shift farther from lower-order $n$-grams over the course of training. These last two findings align with previous work showing that the Kullback–Leibler divergence (Kullback and Leibler, 1951) between Pythia models' output probability distributions and the probability distributions of unigram and bigram models show a similar pattern (Belrose et al., 2024).

Together with the idea that a greater number of parameters increases language models' *capacity* (see, e.g., Wang et al., 2017; Hestness et al., 2017; Kaplan et al., 2020; Allen-Zhu et al., 2019; Allen-Zhu and Li, 2024), these results may suggest that the smaller models may need to rely more on such lower-order $n$-gram-like predictions, while larger models may be able to learn more complex relations between words as they continue to train.

We see two clear patterns with semantic similarity. First, there is little difference between the uniformly-weighted and SGPT-weighted versions of each similarity metric, though the SGPT-weighted variant of the Wikipedia-based fastText word vectors does appear to be consistently more correlated with language model log-probability than the uniformly-weighted variant. There does appear to be a large difference between the two fastText versions, however. Specifically, we see that similarities derived from the Common-Crawl-based vectors have a higher overall correlation to log-probability which peaks at roughly the same time as unigram log-probability does; while the Wikipedia-based metrics have a lower correlation overall which peaks more concurrently with that of trigram log-probability. This likely relates to the correlation between the similarity metrics and unigram log-probability—the Pearson correlation coefficient between the Common-Crawl-based similarities and each metric of unigram log-probability is between 0.67 and 0.69, while the correlations with Wikipedia-based similarities lie between 0.34 and 0.35 (see Appendix H).

Finally, we note that the patterns across random seeds are remarkably similar—the confidence intervals are virtually invisible for the most part. Seed-level analyses (Appendix F) reveal that the larger confidence intervals in the first 10 steps of the Pythia models with PolyPythia seeds (i.e., the 14M–410M models) are due to small differences between models in the early stages of training; while the large confidence interval in the last steps of the Open-GPT2 345M models is driven by a single outlying checkpoint (step 256,000 of the `beren` (seed 49) model). In fact, Parc-Pythia, Parc-Mamba, and Parc-RWKV show a Pearson correlation $r \geq 0.93$ overall at each step $\geq 80$; not only across seeds of the same model, but also across architectures (see Appendix G).

## 4 Experiment 2: Predicting Language Model Log-Probabilities with Heuristics

Experiment 1 revealed patterns in the correlation between simple heuristics and language model log-probability. But to what extent do these heuristics correspond to dissociable facets of language model behavior? On the one hand, all the heuristics are at least weakly correlated (see Appendix H), presenting a possible confound to interpretation. As an example, Common-Crawl-derived semantic similarity is highly correlated with unigram log-probability ($r = 0.67 - 0.69$ depending on weighting) and its correlation with language model log-probability follows the same trajectory. Thus, it is possible that the correlation between Common-Crawl-derived contextual semantic similarity and language model log-probability may be largely explained by the former simply capturing unigram probability (i.e., frequency)—for example, it is plausible that lower-frequency words will have lower-quality word embeddings and occur in fewer contexts, reducing their average similarity to other words. On

the other hand, this is clearly not the case for all heuristics—we see different patterns in the degree of correlation between language model log-probabilities over the course of training and different $n$-gram log-probabilities, as well as a unique pattern for Wikipedia-derived contextual semantic similarity.

Additionally, there are reasons to believe that language models may in fact learn to make predictions that align with the $n$-gram and contextual similarity heuristics implicitly as part of autoregressive language modeling, based on the nature of the task itself. For example, a word that is more common (i.e., that has a higher unigram probability) is by definition more likely to occur again, and similarly, a word that often follows a specific sequence of words (i.e., that has a high $n$-gram probability for $n > 1$) is likely to do so again. In a similar way, in a coherent text, one should generally expect a given word to be close in meaning (i.e., contextually semantically similar) to the words in its preceding context (for discussion, see Michaelov and Bergen, 2022; Michaelov et al., 2023, 2025). Indeed, a sensitivity to contextual semantic similarity may at least partly explain how language models are able to make predictions that align with real-world knowledge (as in, e.g., Zellers et al., 2018, 2019; Forbes et al., 2019; Bisk et al., 2020; Sakaguchi et al., 2020; Jones et al., 2022; Kauf et al., 2023). There is also mechanistic evidence of transformers directly implementing $n$-gram prediction (Bietti et al., 2023; Wang et al., 2025; Chang and Bergen, 2025); and the fact that transformers directly learn associations between tokens (e.g., Meng et al., 2022; Bietti et al., 2023; Nichani et al., 2024) and appear to be able to make predictions at the concept as well as token level (Feucht et al., 2025) suggests that prediction based on contextual semantic similarity is something that could in principle be learned.

In this experiment, we focus on a precise characterization of language model behavior in terms of how it relates to the aforementioned heuristics. First, we revisit the finding in Section 3 that over the course of training, language model log-probabilities correlate better with $n$-gram log-probabilities of increasing order, and correspondingly correlate worse with the log-probabilities of lower-order $n$-grams. We ask whether language model predictions are still biased toward lower-order $n$-gram probabilities even as they begin to reflect higher-order $n$-grams. Specifically, we test whether language model log-probability is sensitive to unigram log-probability above and beyond the extent to which this is captured in 5-gram log-probability. Second, we ask whether language model predictions are sensitive to contextual semantic similarity above and beyond that implicitly captured by unigram and 5-gram probability. This is crucial because, as previously stated, unigram probability may impact contextual semantic similarity metrics due to its effect on the word vectors learned; and, as stated, coherent sentences are likely to involve words that are semantically similar to their context, and thus, if these sequences are learned, they may implicitly contain contextual semantic similarity information.

As far as we are aware, these questions are novel in language model research. But they have been studied in the context of human language processing. Specifically, a number of studies have investigated whether there are dissociable effects of word frequency, contextual probability (derived from $n$-grams or neural language models), and semantic similarity on neural and behavioral indices of prediction in humans, with varying results (Lau et al., 2013; Frank and Willems, 2017; Nieuwland et al., 2020; Shain, 2024; Michaelov et al., 2024; Opedal et al., 2024). We follow the approach taken in these studies, using the $n$-gram log-probabilities and similarity metrics to predict language model log-probability with a linear regression, allowing us to estimate the extent to which language models show a bias in prediction toward each heuristic while controlling for the possible influence of the others.

## 4.1   Method and Analysis

The evaluation dataset, cosine similarities, and $n$-gram and language model log-probabilities were all the same as in Experiment 1. Instead of computing raw correlations, we instead fit linear regressions that predict the dependent variable—in this case, language model log-probability—based on all predictors of interest, namely, unigram log-probability, 5-gram log-probability, and contextual semantic similarity. Thus, as in human studies, we are able to investigate whether correlations between unigram language model log probability and our predictor variables are dissociable.

We are also interested in the robustness of these patterns. Thus, in addition to using only $n$-grams calculated from the same corpus the language model was trained on (i.e., either OpenWebText or The Pile), we also carry out the same analysis with the $n$-grams based on the other corpus. For the previously-discussed reasons, we are also interested in whether there is a difference between the two similarity metrics, and so we also construct regressions with each. Thus, all regressions

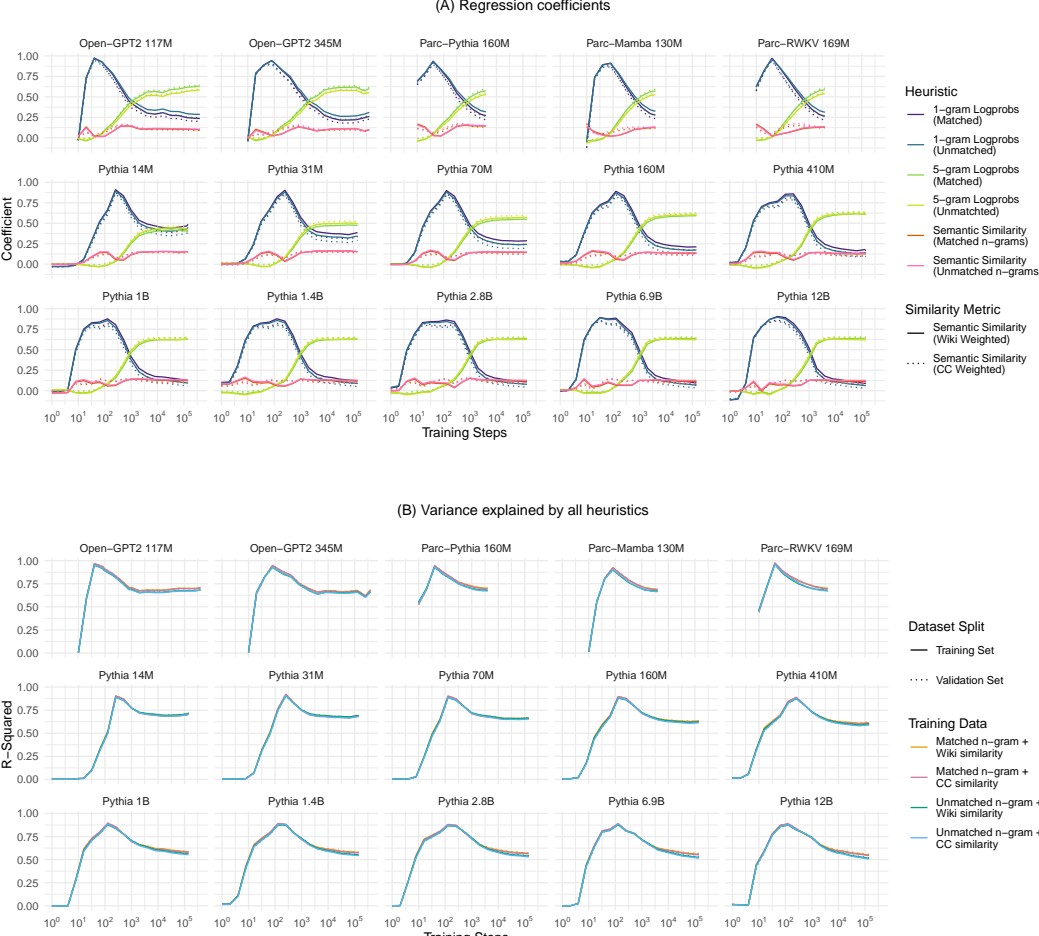

Figure 2: (A) Regression coefficients of the three heuristics over the course of training under different conditions, specifically, whether the $n$-gram data is the same as that on which the language model was trained (matched) or not (unmatched), and whether SGPT-weighted contextual semantic similarity metric is calculated using Common-Crawl-based or Wikipedia-based fastText word vectors.
(B) Proportion of the variance in language model log-probability explained by the regressions in Figure 2. We also report the $R^2$ values of the same regressions' predictions on the validation set.

include 3 predictors: unigram log-probability (OpenWebText or The Pile), 5-gram log-probability (OpenWebText or The Pile; the same corpus as unigram log-probability), and semantic similarity (Wikipedia-based or Common-Crawl-based). In order to more easily compare the coefficients, we $z$-transform all variables in the regression (we provide the coefficients for the same regressions with un-normalized variables in Appendix K).

## 4.2 Results and Discussion

We present our results for all regressions with SGPT-weighted contextual semantic similarity in Figure 2. We find that as in Experiment 1, while the exact steps at which the phases themselves occur differs somewhat across models, the phases themselves are remarkably clear and consistent.

In **Phase 1**, the coefficient of unigram log-probability on language model log-probability increases sharply from zero (or near-zero). This increase peaks at the same time as the peak in Pearson correlation seen in Figure 1. Concurrently, the coefficient of semantic similarity (both metrics) increases. By contrast, there is no positive increase in the coefficient of the 5-gram log-probability—in fact, it slightly decreases below zero during this phase.

In **Phase 2**, the coefficient of unigram log-probability decreases and the coefficient of 5-gram probability increases. There is also a sharp increase in the coefficient of 5-gram log-probability. At or soon after the beginning of this phase is a small trough in the coefficient of Wikipedia-derived similarity, which is smaller (or nonexistent) for Common-Crawl-derived similarity.

In **Phase 3**, the change in coefficients slows down and begins to stabilize for all models.

These patterns are consistent across all models, with the main difference being in the timing of the phases. In the Pythia models, phases generally occur earlier in the larger models than in the smaller models, but in the two Open-GPT2 models, this pattern is reversed. It notably does not appear that training tokens explain the timings of the phases better than training steps—in fact, despite being trained on $\sim 0.5$M tokens per step, the models trained on OpenWebText (Open-GPT2, Parc-Pythia, Parc-RWKV, and Parc-Mamba) generally enter phases in fewer steps than the Pythia models which are trained on $\sim 2$M tokens per step. Model size, however, does appear to impact coefficient size—smaller models see a smaller drop in the coefficient of unigram log-probability and a smller increase in the coefficient of 5-gram log-probability than larger models. There is also some variability in the peaks and troughs of the coefficients of the semantic similarity metrics, though they all appear to follow the general trajectory described above. To further verify the robustness of these patterns, we also look at the seed-level results for regressions including both SGPT-weighted and unweighted variants of the similarity metrics (provided in Appendix I), which produce virtually identical patterns of coefficients.

We next turn to the overall fit of the regressions to language model log-probability. The $R^2$ of each regression provides an estimate of the proportion of the variance in language model log-probability explained by the regression. Overall fit is largely shaped by the strength of the effect of unigram log-probability—the regressions best predict language model log-probability ($R^2 = 0.86 - 0.98$ depending on the model) when the effect of unigram log-probability on language model log-probability is at its peak, after which fit decreases sharply as the strength of the effect of unigram log-probability decreases sharply, and levels off as the decrease in the strength of unigram log-probability slows down. However, the overall fit still remains relatively high even after this decrease—it stabilizes for the models with fewer than $\sim 0.5$B parameters and does not fall below $R^2 = 0.5$, even for the largest models. In fact, except for the earliest steps of training (Pythia 14M: step $\leq 128$; Pythia 31M: step $\leq 64$; all other models: step $\leq 32$), the heuristics always explain at least half of the variance in model log-probability. As can be seen, the choice of $n$-gram corpus and fastText training corpus do not dramatically change any of these patterns, which are also relatively consistent across seeds (see Appendix J). This suggests that overall fit is robust to these factors. To further evaluate the robustness of our regressions to overfitting, we calculate the $R^2$ on the held-out validation set using predictions from the regressions. Again, we see almost no difference, suggesting that these results are robust. We again see virtually identical patterns when we look at seed-level results for both the SGPT-weighted and unweighted similarity variants (see Appendix J).

## 5  General Discussion

In line with the results of previous work (Karpathy et al., 2016; Choshen et al., 2022; Chang and Bergen, 2022; Chang et al., 2024), we show in Experiment 1 that the extent to which language model predictions correlate with $n$-gram predictions varies over time such that language models trained on more data make predictions that correlate more strongly with higher-order $n$-grams. We show that for $n \in \{1, 2, 3, 4, 5\}$, this applies to transformers across 3 orders of magnitude (14M to 12B parameters), as well as language models of both the RWKV and Mamba architecture. We also find a predictable impact of model scale: while smaller models eventually make predictions that are as correlated (or almost as correlated) with higher-order $n$-grams as the larger models, they retain a higher correlation with the lower-order $n$-grams than the larger models. As the larger models' predictions increasingly correlate with the higher-order $n$-grams, their correlation with the lower-order $n$-grams decreases. This suggests that the larger models shift to predicting in line with higher-order $n$-grams, while the predictions of the smaller models stay closer to those of the lower-order $n$-grams.

Experiment 2 confirmed this through multiple linear regression—unigram log-probability maintains a larger coefficient during the later phases in smaller than larger models, while the reverse is true of 5-gram log-probability. This means that higher-order $n$-grams account more for the behavior of larger models than they do for smaller models. This could be taken to suggest that the smaller models

may rely on lower-order $n$-gram relations—perhaps due to limited capacity—while larger models may be able to more effectively make use of longer contexts. Additionally, the fact that the overall fit of the heuristics decreases the most for the largest Pythia models in the last phase of pretraining combined with the fact that they already begin to perform better than smaller models at standard benchmarks during this phase (Appendix L) suggests that the predictions of these models are likely to reflect other more complex cues at this stage of training. Thus, our results seem to suggest that while language model predictions will (and perhaps must) show a high degree of correlation with the heuristics during training, it is only once they pass this stage that that they begin to perform well on downstream tasks. Indeed, while the exact relationship between the two is not known, research has shown that in-context $n$-gram heads only occur in a model after it has learned to predict $n$-grams in the training data (Bietti et al., 2023; Wang et al., 2025). Crucially, in-context $n$-gram heads such as induction heads are thought to greatly contribute to language models' in-context learning capabilities (Elhage et al., 2021; Olsson et al., 2022; Bietti et al., 2023; Akyürek et al., 2024; Edelman et al., 2024; Chen et al., 2024b; Crosbie and Shutova, 2025), which in turn have been argued to be responsible for a wide range of language model behaviors, including performance on downstream tasks (see, e.g., Lampinen et al., 2025). Despite this, there is also evidence that the tendency for language models to make predictions that align with these heuristics persists throughout the course of training. In fact, it has been shown that as language models get larger and are trained on more data, they continue to (and may even increase the extent to which they do) predict high-probability $n$-grams (McKenzie et al., 2023; Michaelov and Bergen, 2023) and words that are semantically related to their context (Michaelov et al., 2025; Gonen et al., 2025), even when they should not. An interesting line of future work would be to investigate whether it is possible to predict the extent to which language models are susceptible to these kinds of phenomena based on analyses such as those carried out in Experiment 2.

Are the predictions of larger models trained on more data biased toward lower-order $n$-grams above and beyond the extent to which they are accounted for by higher-order $n$-grams? On the one hand, decreases in the size of the 1-gram coefficient are usually matched by increases in the size of the 5-gram coefficient, and vice-versa; suggesting some degree of competition or incompatibility between the two. On the other hand, even as they become increasingly sensitive to higher-order $n$-gram probabilities, lower-order $n$-gram probabilities (or at least, unigram probabilities) are still overweighted in language model predictions; that is, the models over-predict words with high unigram probability relative to their 5-gram probability. This is in line with work showing that lower-order $n$-gram circuits—specifically, bigram circuits—persist over the course of training in transformer models, though they appear to become less effective and less distinct in later stages (Chang and Bergen, 2025). Whether this is also true for the RWKV and Mamba models is a question for future work.

Previous work has demonstrated a relationship between the probabilities assigned by language models to words in context and the semantic similarity between those words and their contexts (Michaelov et al., 2021, 2024). In this paper, we provide what is, as far as we are aware, the first investigation of both the extent to which this correlates with predictions above and beyond other factors thought to impact them (in this case, $n$-gram probability), and the extent to which this varies over the course of training. We find that while the direct correlation between semantic similarity and language model log-probability varies depending on how semantic similarity is calculated, after accounting for unigram and 5-gram log-probability, most of this difference disappears—we see the correlation with similarity emerge relatively early, and remain throughout training.

Additionally, we find that while the weighting method used to create the context vector—uniform vs. SGPT-weighting—does not appear to have much of an impact, there was a difference between the two different sets of vectors we used. Specifically, the Common-Crawl-based fastText vectors (Grave et al., 2018) are more closely correlated with unigram probability than the Wikipedia-based vectors (Bojanowski et al., 2017), which could explain the differences we see in their relationship to language model log-probability. What explains the difference between these embedding vectors? One intuitive possibility is that the difference is due to differences in training corpus. However, another plausible explanation is that it relates to how the embeddings were trained. Specifically, the Wikipedia embeddings were trained using a skip-gram model (Bojanowski et al., 2017), and the Common Crawl vectors were trained using a continuous bag-of-words (CBOW) model (Grave et al., 2018). Previous work has suggested that skip-gram models learn higher-quality representations for infrequent words (Mikolov et al., 2013), and thus, a likely reason for the higher correlation between

the CBOW-based embeddings and unigram frequency is simply that lower-frequency words have lower-quality representations, and thus their similarity to the context is under-estimated.

Our results have several key implications. First, they provide evidence that across architectures and scales, there is a consistent relationship between the predictions of language models and $n$-gram probability. A strong interpretation of our results, combined with similar results on recurrent neural networks by Karpathy et al. (2016), is that a system engaged in autoregressive language modeling will inevitably go through phases of overfitting to $n$-grams of increasing $n$ before learning more abstract patterns. We also for the first time observe a consistent correlation between semantic similarity and language model predictions over the course of training, something that can be detected even after accounting for any implicit semantic similarity relations inherent in $n$-grams (i.e., based on co-occurrence). Finally, while it has previously been demonstrated that it is possible to construct a rule based on $n$-gram statistics that can explain a given language model prediction (Nguyen, 2024), we see that, except for in the very early stages of training, there is a specific weighting of three simple heuristics (plus a constant) that can explain over half of the variance in the predictions made by any seed of any language model tested at any point during training.

# 6 Limitations

While our results are consistent across our models, our analysis is limited to three architectures, and in the case of Mamba and RWKV, only relatively small models (130–169M models trained on ∼2B tokens). Thus, it is possible that other models could show different training dynamics. We also limit our analysis to $n$-grams in $n \in \{1, 2, 3, 4, 5\}$ and static word embeddings, but language models may be sensitive to higher-order $n$-grams and semantic similarity that is better reflected by contextual word embeddings or text embeddings. Finally, our regressions still do not account for all the variance in language model behavior—there is still a lot to be understood, even in the smallest models.

# 7 Conclusions

Language model behavior throughout training can largely be accounted for by several simple heuristics: word frequency, $n$-gram probability, and semantic similarity. The extent to which each of these correlates with language model predictions has a consistent pattern across models of different sizes, models trained on different data, and models with different architectures. Taken together, this suggests that the autoregressive language modeling task itself may be the largest factor—and perhaps the decisive one—in shaping the behavioral phases that language models pass through. It may be that models cannot help but first crawl through $n$-gram predictions in their gradient descent toward mature next-word prediction.

# Acknowledgments

We would like to thank the members of the Computational Psycholinguistics Laboratory at MIT and the Language and Cognition Laboratory at UCSD for their valuable advice and discussion. James Michaelov was supported by a grant from the Andrew W. Mellon foundation (#2210-13947) during the writing of this paper.

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

# A  Language Model Details

## A.1  Pretrained Models

We summarize the details of all models analyzed in Table 1. Further details of each set of models are discussed in the relevant sections below. scc

Table 1: Details of the models used in our analyses.

| Model | Parameters | Seeds | Tok/Step | Steps |
|---|---|---|---|---|
| Pythia | 14M | 10 | | |
| | 31M | 10 | | |
| | 70M | 10 | | |
| | 160M | 10 | | |
| | 410M | 10 | | 0, 1, 2, 4, 8, 16, 32, 64, 128, 256, 512, |
| | 410M | 10 | 2M | 1000, 2000, 4000, 8000, 16000, |
| | 1B | 1 | | 32000, 64000, 128000, 143000 |
| | 1.4B | 1 | | |
| | 2.8B | 1 | | |
| | 6.9B | 1 | | |
| | 12B | 1 | | |
| Open-GPT2 | 117M | 4 | 0.5M | 0, 10, 20, 40, 80, 100, 200, 400, 800, 1000, 2000, 4000, 8000, 16000, 32000, 64000, 128000, 256000, 400000 |
| | 345M | 4 | | |
| Parc-Pythia | 160M | 6 | | 10, 20, 40, 80, 160, 320, 640, 1280, |
| Parc-RWKV | 169M | 6 | 0.5M | 2560, 4000 |
| Parc-Mamba | 130M | 6 | | |

**Pythia**  The Pythia (Biderman et al., 2023b) models are a set of autoregressive language models based on the GPT-Neo models (Black et al., 2021; Andonian et al., 2021), which were created as an attempt to train fully-open versions of the smaller GPT-3 language models (Brown et al., 2020). Thus, all models are trained on a 300B-token dataset (The Pile; Gao et al., 2020). Our analysis includes the main (seed 1234) 14M, 31M, 70M, 160M, 410M, 1B, 1.4B, 2.8B, 6.9B, and 12B parameter models. In addition, we use the recently-released 14M, 31M, 70M, 160M, 410M PolyPythia models (van der Wal et al., 2024) trained on a further 9 random seeds each (1, 2, 3, 4, 5, 6, 7, 8, 9); thus, we have 10 models for each of these parameter sizes. With all Pythia models, we calculated word probabilities with model checkpoints at the following steps: 0, 1, 2, 4, 8, 16, 32, 64, 128, 256, 512, 1000, 2000, 4000, 8000, 16000, 32000, 64000, 128000, and 143000 (fully-trained). Note that models were trained on 2M tokens at each step. Because of a known issue with the `fp16` precision used during training (see Liang, 2024), we run all models at `fp32` precision.

**Open-GPT2**  We also carry out our analyses on the GPT-2 models trained as part of the Mistral project (Karamcheti et al., 2021). These are a set of GPT-2 small (117M parameters) and GPT-2 medium (345M parameters) language models trained on the OpenWebText (OWT) corpus (Gokaslan and Cohen, 2019), with training checkpoints released. For each size, models trained with 5 random seeds were released. Due to a known issue with the GPT-2 Medium model with random seed 21 (see Hawkins, 2023), we used the 4 remaining random seeds of each model (namely, seeds 49, 81, 343, and 777). For each of the models trained with each of these random seeds, we calculated language model probabilities at checkpoints corresponding to the following steps: 0, 10, 20, 40, 80, 100, 200, 400, 800, 1000, 2000, 4000, 8000, 16000, 32000, 64000, 128000, 256000, and 400000 (fully trained). Note that models were trained on 0.5M tokens at each step.

## A.2  Parc Models

In order to investigate the effect of architecture, we also train our own models. Specifically, we train Pythia (Biderman et al., 2023b), Mamba-1 (Gu and Dao, 2024), RWKV-4 (Peng et al., 2023) models.

We selected one model of each architecture of comparable size—namely, Pythia 160M, RWKV 169M, and Mamba 130M. These models were adjusted these models to all use the Pythia tokenizer, and then trained from scratch (i.e., random initialization) on the OpenWebText corpus for 4000 steps, where each step consisted of a 1024-token length sequence with a batch size of 512, for a total of 0.5M tokens, following Karamcheti et al. (2021). All hyperparameters were chosen to match those of the original model training where possible (Biderman et al., 2023b; Peng et al., 2023; Gu and Dao, 2024). We provide the full training code.

For each of the three models selected, we trained six models with different random seeds (0, 1, 2, 3, 4, 5), each of which seed was applied both to the data collator and initialization. We analyze data from steps 10, 20, 40, 80, 160, 320, 640, 1280, 2560, and 4000 (fully-trained).

## B Estimating $n$-gram probabilities using infini-gram

We estimate the $n$-gram probability $\hat{p}$ of a word $w_i$ using the *Stupid Backoff* scheme (Brants et al., 2007) as described in Equation 1, where, following Brants et al. (2007), we set $\alpha = 0.4$.

$$\hat{p}(w_i|w_{i-n+1}^{i-1}) = \begin{cases} \frac{c(w_{i-n+1}^i)}{c(w_{i-n+1}^{i-1})}, & \text{if } c(w_{i-n+1}^i) > 0 \\ \alpha \hat{p}(w_i|w_{i-n+1}^{i-1}), & \text{otherwise} \end{cases} \tag{1}$$

To calculate unigram probability, or in cases where the recursive process described in Equation 1 reaches $n = 1$, also following Brants et al. (2007), we calculate $\hat{p}(w_i)$ as described in Equation 2, where $|C|$ describes the total number of tokens in a corpus, as provided by *infini-gram* (Liu et al., 2024).

$$\hat{p}(w_i) = \frac{\max\{1, c(w_i)\}}{|C|} \tag{2}$$

As Brants et al. (2007) note, this approach does not provide true probabilities that sum to 1, but these estimates perform similarly well to Kneser-Ney-smoothed (Kneser and Ney, 1995) true $n$-grams, especially on datasets of the scale used in the present work. Additionally, we note that for the unigram counts, we use the word-level counts in the numerator but the token-level counts in the denominator, and thus, unigram probability may be systematically under-estimated relative to other $n$-grams. Finally, because our larger $n$-grams are calculated at the word level rather than the token level, we expect that our findings may slightly diverge from mechanistic interpretability work looking at token-level relationships (Voita et al., 2024; Chang and Bergen, 2025). However, due to the large size of all the datasets involved, it is unlikely that these factors would drastically impact any of the overall patterns identified in this work.

## C Contextual Similarity

Given a sequence of words $w_1...w_i$, the contextual semantic similarity between $w_i$ and its context $c = w_1...w_{i-1}$ is defined as the cosine similarity between the $\vec{w_i}$, the word embedding vector of $w_i$, and $\vec{c}$, the embedding of the context. A word embedding vector $\vec{w_i}$ is simple the embedding of a word as provided using the *fastText* package (Bojanowski et al., 2017). In our study, we use both the Wikipedia-derived fastText vectors of Bojanowski et al. (2017) and the Common-Crawl-derived vectors of Grave et al. (2018). $\vec{c}$ is calculated as in Equation (3):

$$\vec{c} = \sum_{j=1}^{j=i-1} \beta_j \vec{w_j} \tag{3}$$

With uniform weighting, $\beta = \frac{1}{i-1}$. With SGPT weighting (Muennighoff, 2022), $\beta = \frac{j}{\sum_{k=1}^{k=i-1} k}$.

## D   Evaluation dataset

In order to evaluate how well our heuristics predict language model behavior, we construct a dataset made up of Natural Words in Context (NaWoCo), in sequences previously unseen by the models. We describe the dataset construction process below.

First, we extracted 250,000 sentences from the FineWeb corpus (Penedo et al., 2024) that had more than 5 words, began with a capitalized word, had no other capitalized words, were assigned a probability of being toxic of 0.1 or less (using the model released by Logacheva et al., 2022), and are not in the training data of any of the language models tested—that is, they were not in the OpenWebText (Gokaslan and Cohen, 2019) or Pile (Gao et al., 2020) corpora as determined using infini-gram counts. This last step was to ensure no overlap between these sentences and the training data of the models, which could impact model behavior.

From this sample, we then selected 100,000 unique sentences for the training set, and 50,000 each for the validation and test sets. Because we are interested in the probabilities of words in context, we then randomly selected one word (that occurred as the fifth word or later) in each sentence as the *critical word*, that is, the word for which all metrics would be calculated. We then again filtered all these truncated sentences such that they were unique, not toxic, and did not occur in either of the training corpora using the same previously-described method. Finally, we filtered our dataset to ensure that all words were single tokens in the vocabularies of all the language models.

The final NaWoCo dataset is thus made up of a training set of 77,999 items, a validation set of 39,474 items, and a test set of 40,980 items.

## E   Spearman Correlations Between Language Model Log-Probabilities and Predictors

We provide the model-level Spearman correlations between the heuristics and language model log-probability (Figure 3).(Figure 5).

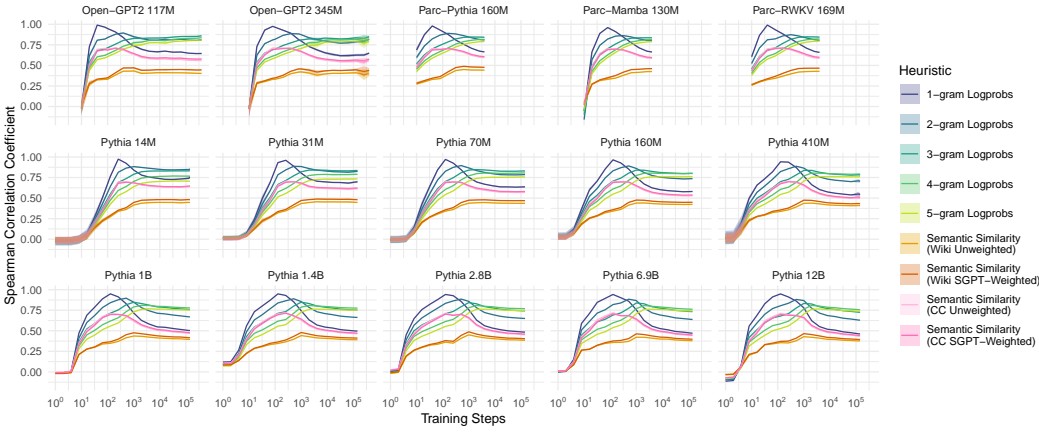

Figure 3: Spearman correlation coefficient $\rho$ between language model log-probability and heuristic metrics ($n$-gram log-probability and word embedding cosine similarity). We show the mean values for all models across seeds and their 95% confidence intervals.

## F   Seed-Level Correlations Between Language Model Log-Probabilities and Predictors

We provide the seed-level Pearson (Figure 4) and Spearman (Figure 5) correlations between the heuristics and language model log-probability for all language models.

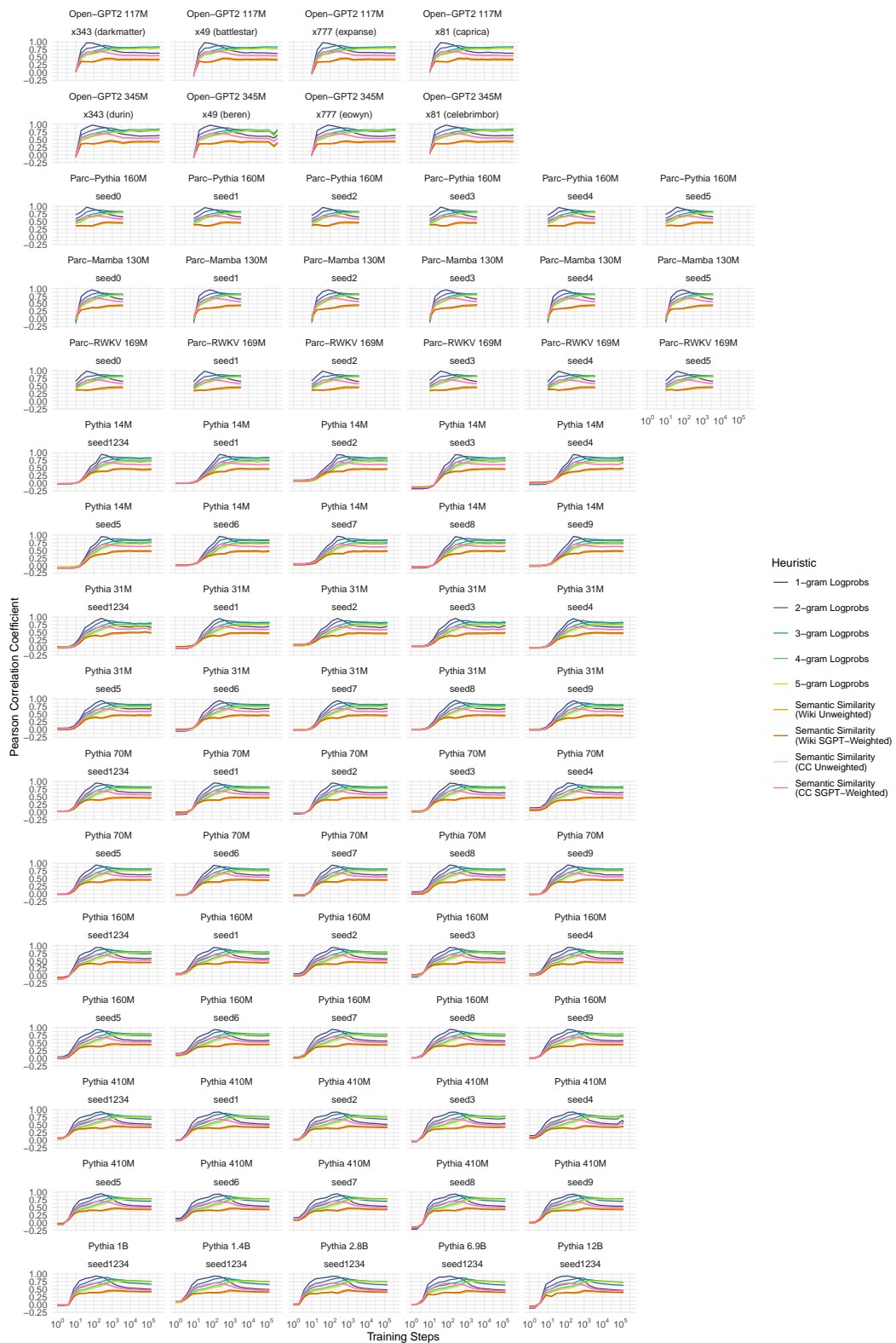

Figure 4: Seed-level Pearson correlation coefficient $r$ between language model log-probability and heuristic metrics ($n$-gram log-probability and word embedding cosine similarity).

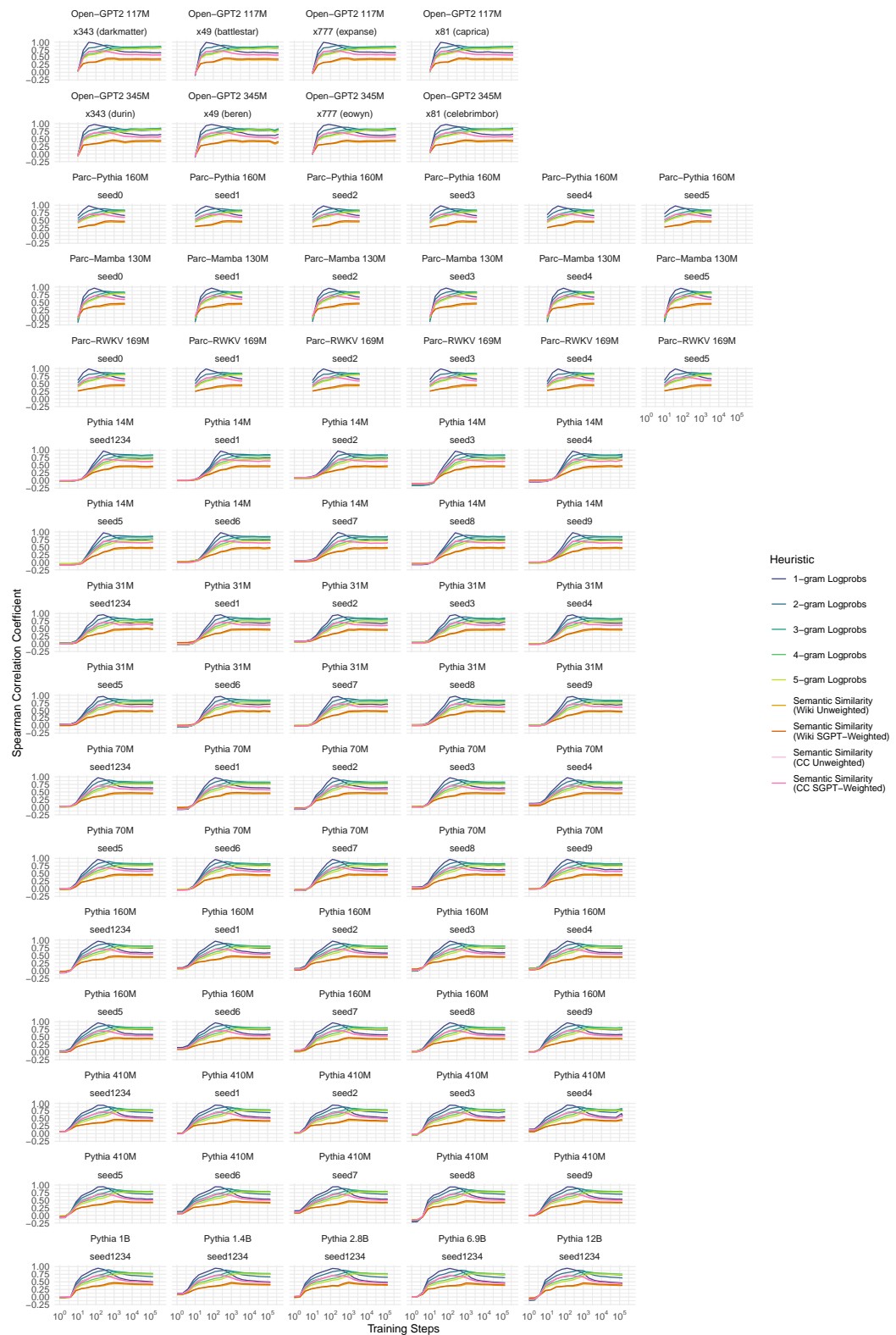

Figure 5: Spearman correlation coefficient $\rho$ between language model log-probability and heuristic metrics ($n$-gram log-probability and word embedding cosine similarity) at the seed level.

# G  Cross-Architecture Similarity

We provide the the Pearson correlation $r$ between each seed of each of the models we train (i.e., Parc-Pythia, Parc-Mamba, and Parc-RWKV) at each time step.

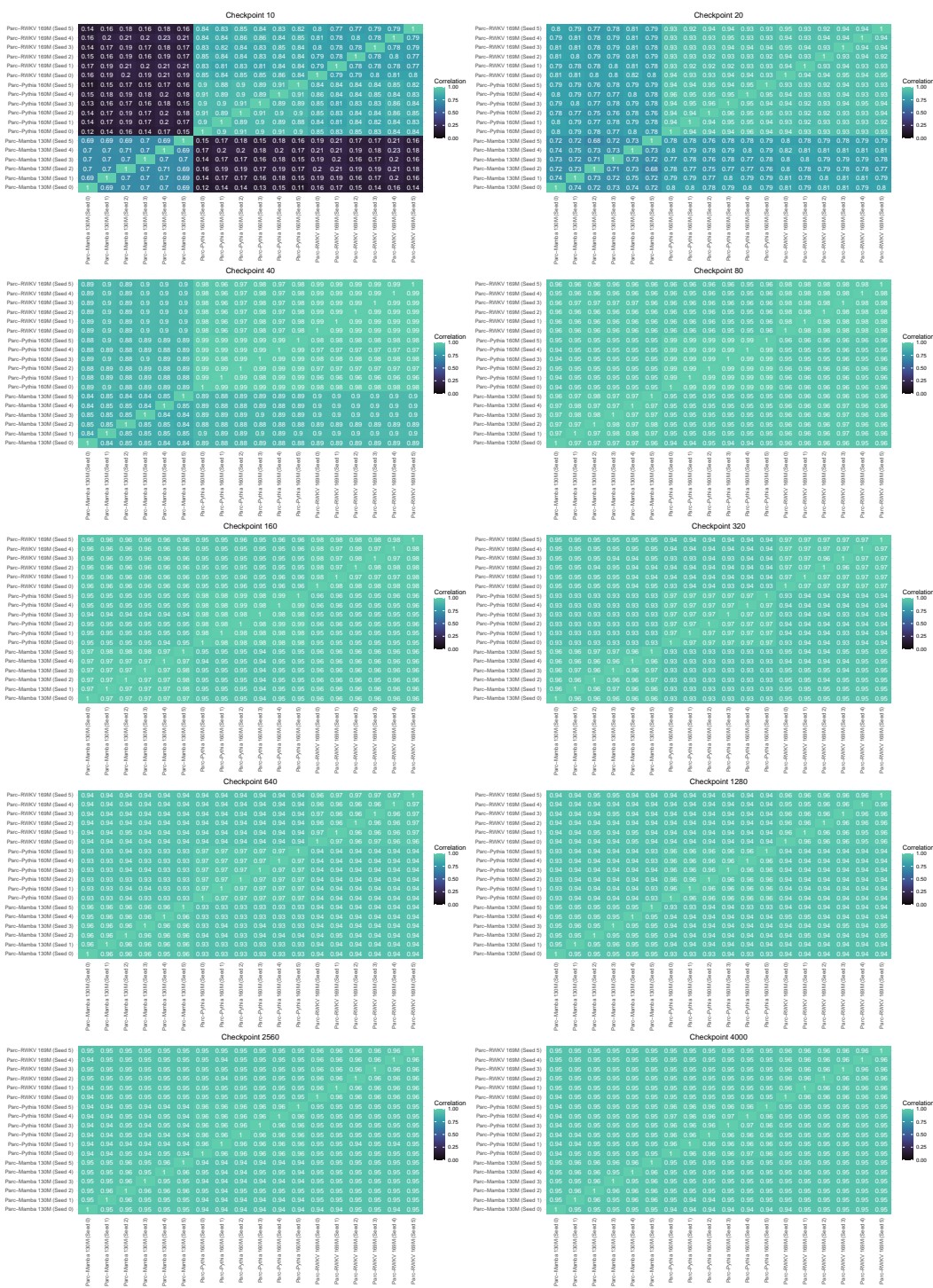

Figure 6: Pearson correlation $r$ between log-probabilities calculated between each model.

# H Correlations Between Predictors

Figure 7 shows the Pearson correlation coefficient $r$ between all heuristics.

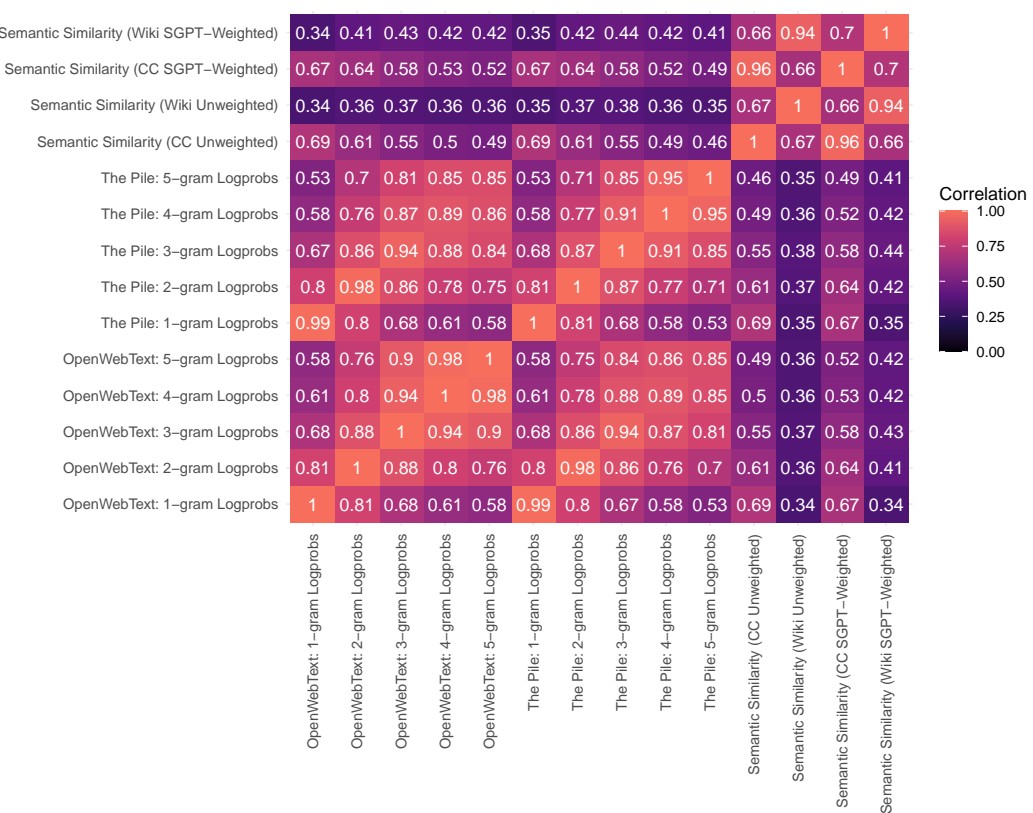

Figure 7: Pearson correlation coefficient $r$ between all predictor variables.

# I Seed-level Coefficients

We provide the coefficents predicted for each seed of each model for each semantic similarity configuration, namely, SGPT-weighted Wikipedia-based similarity (Figure 8), unweighted Wikipedia-based similarity (Figure 9), SGPT-weighted Common-Crawl-based similarity (Figure 10), and unweighted Common-Crawl-based similarity (Figure 11).

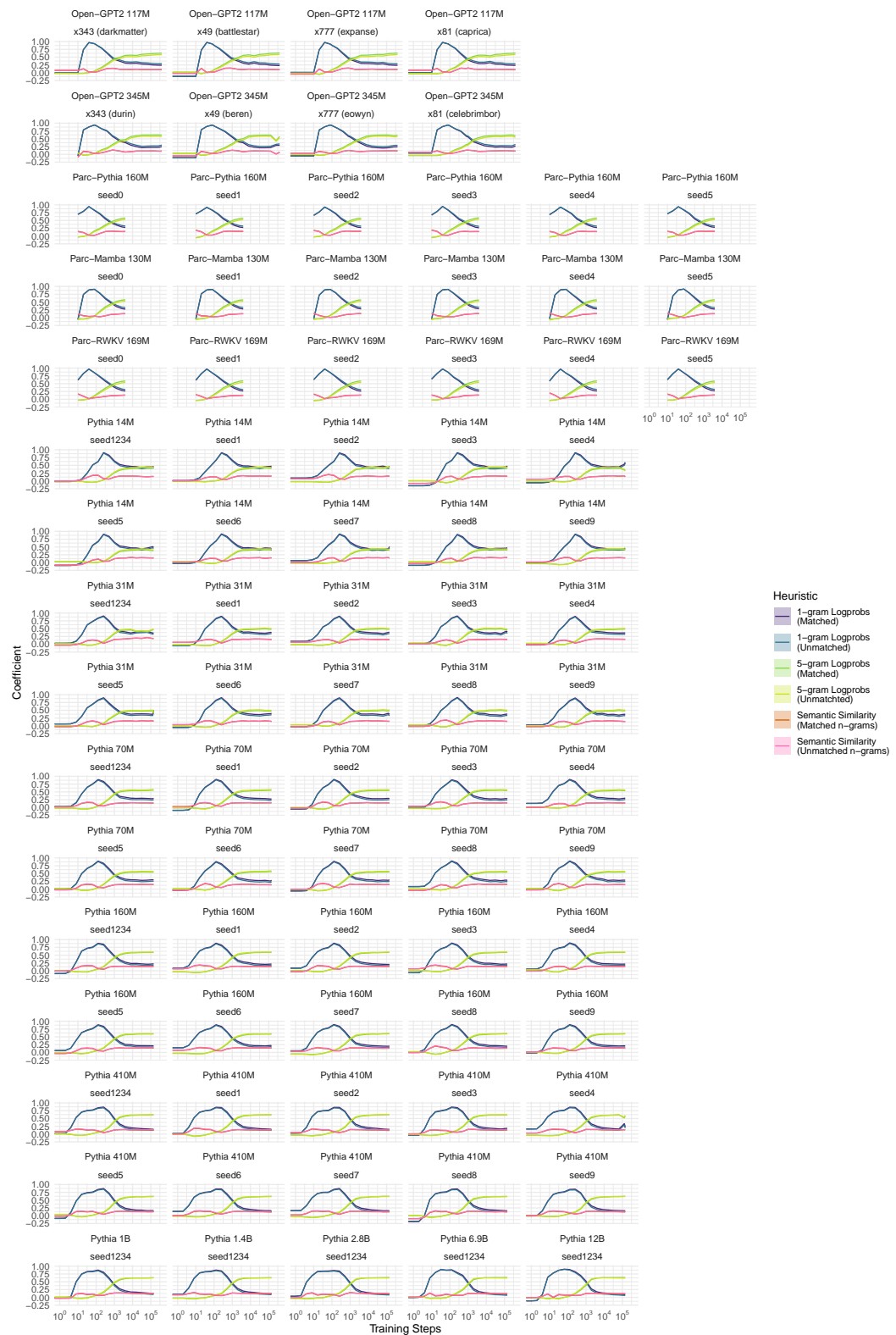

Figure 8: Seed-level coefficients for SGPT-weighted Wikipedia-based similarity, with 95% confidence intervals

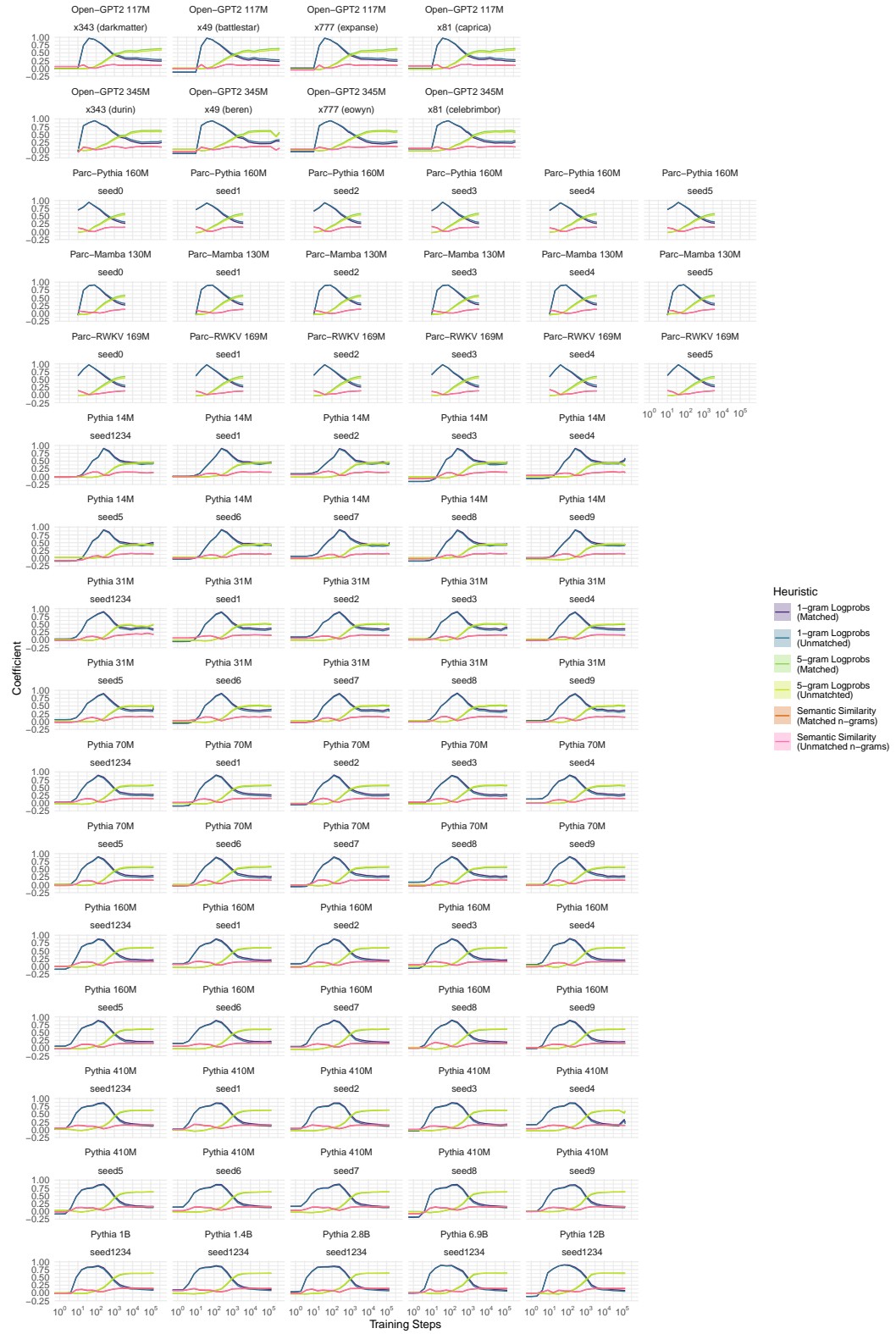

Figure 9: Seed-level coefficients for unweighted Wikipedia-based similarity, with 95% confidence intervals

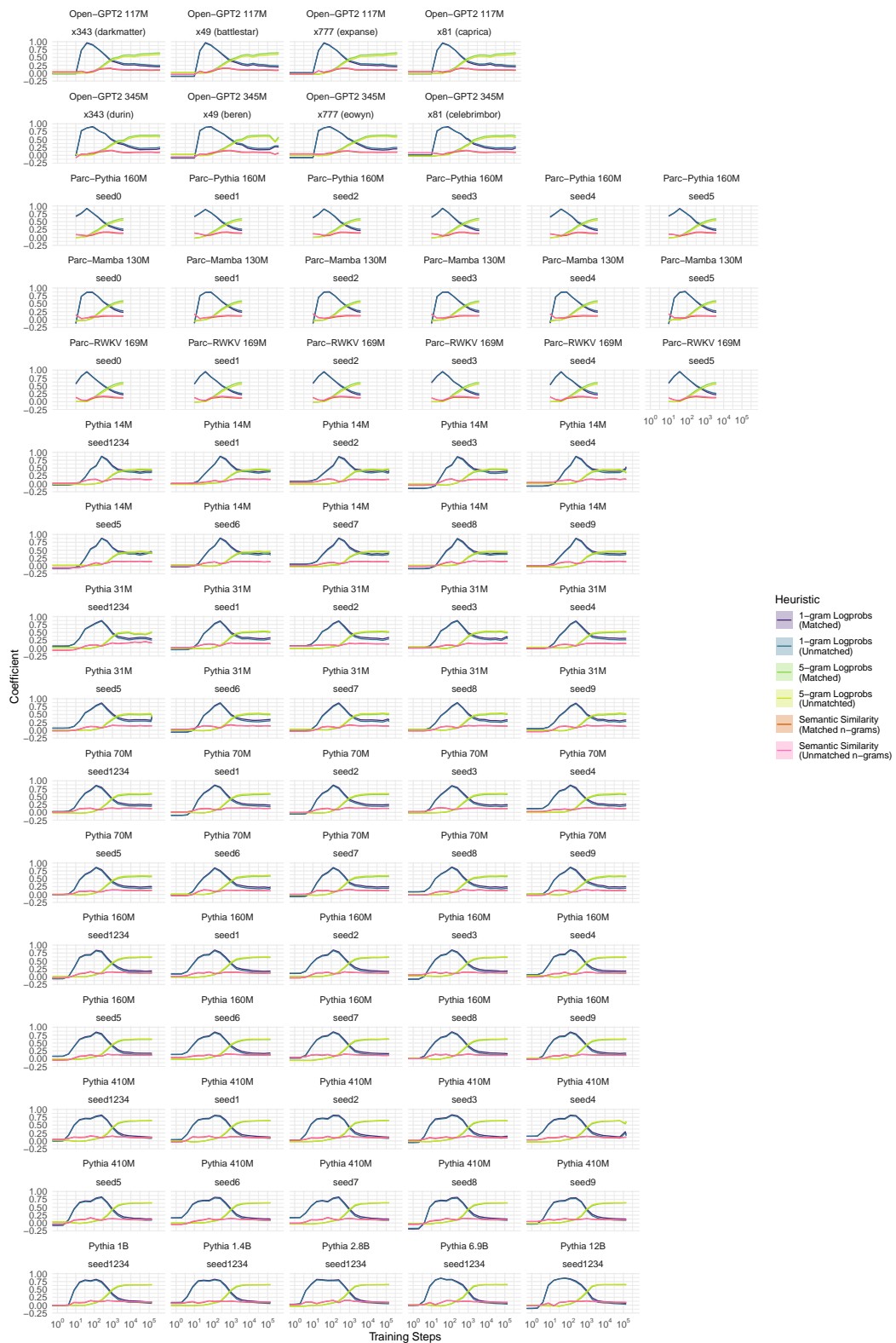

Figure 10: Seed-level coefficients for SGPT-weighted Common-Crawl-based similarity, with 95% confidence intervals

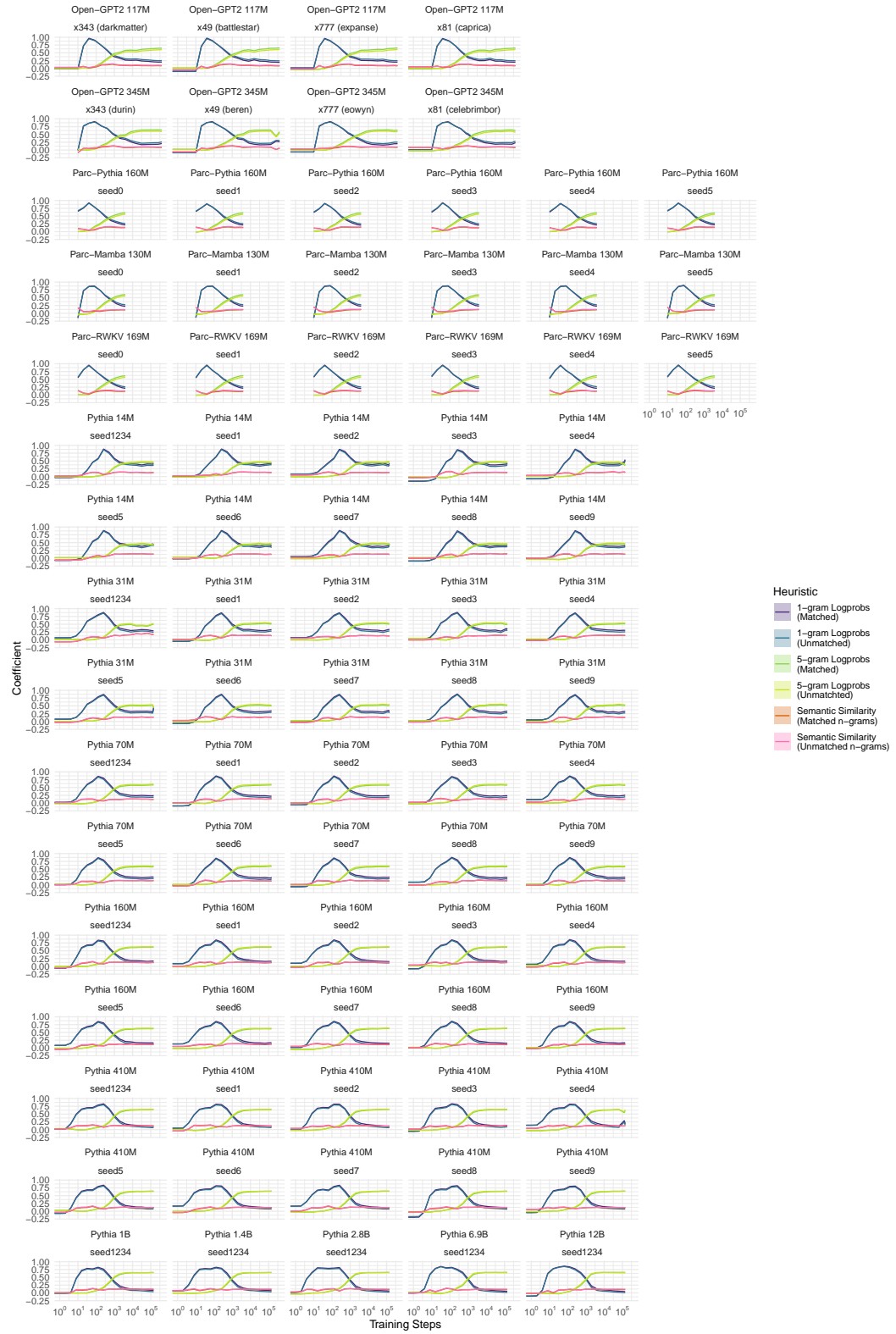

Figure 11: Seed-level coefficients for unweighted Common-Crawl-based similarity, with 95% confidence intervals

## J  Further $R^2$ analyses

We provide plots illustrating the differences in regression $R^2$ across seeds for regressions including SGPT-weighted (Figure 12) and uniformly-weighted (Figure 13) similarity.

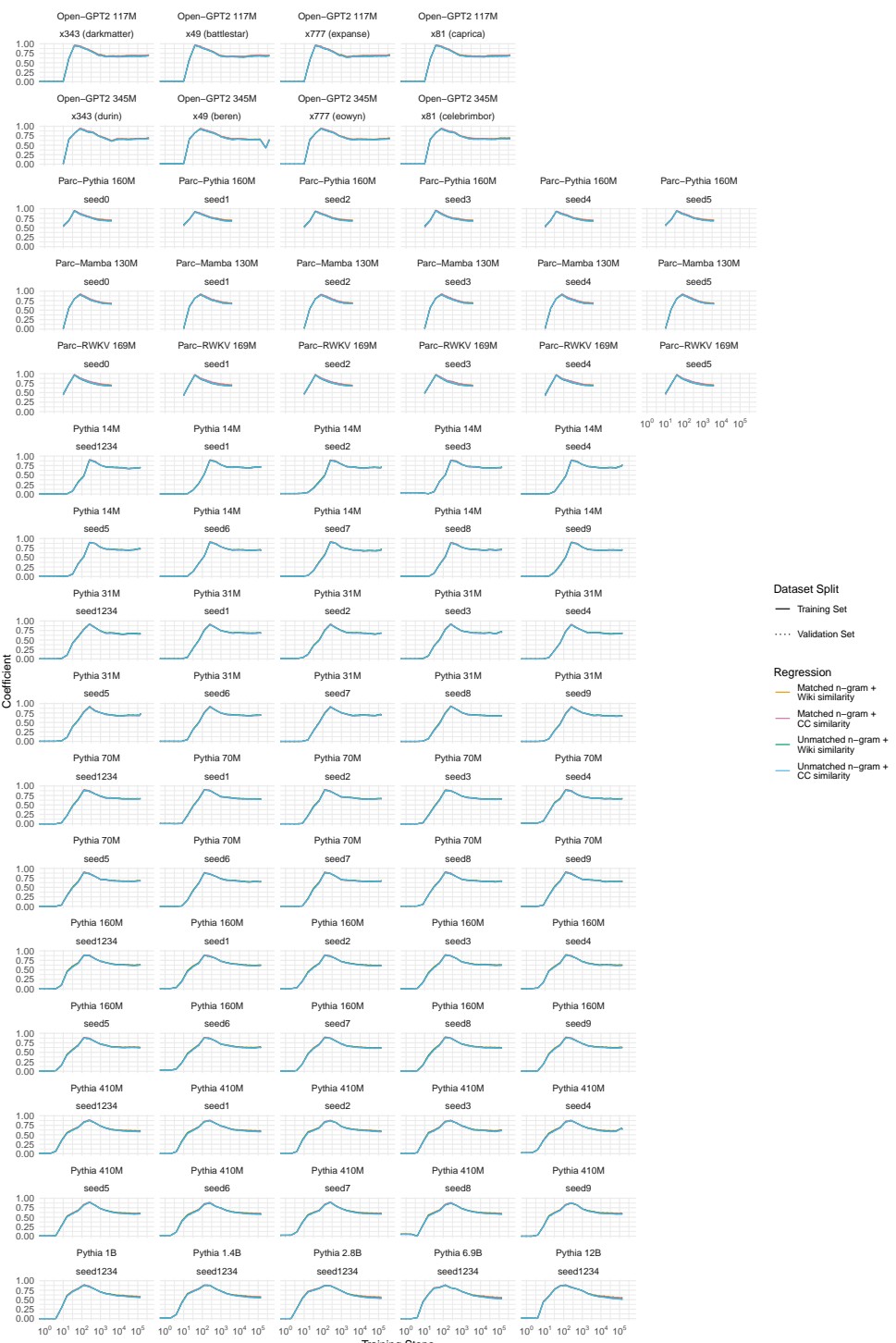

Figure 12: Proportion of the variance in language model log-probability explained by the regressions including SGPT-weighted similarity.

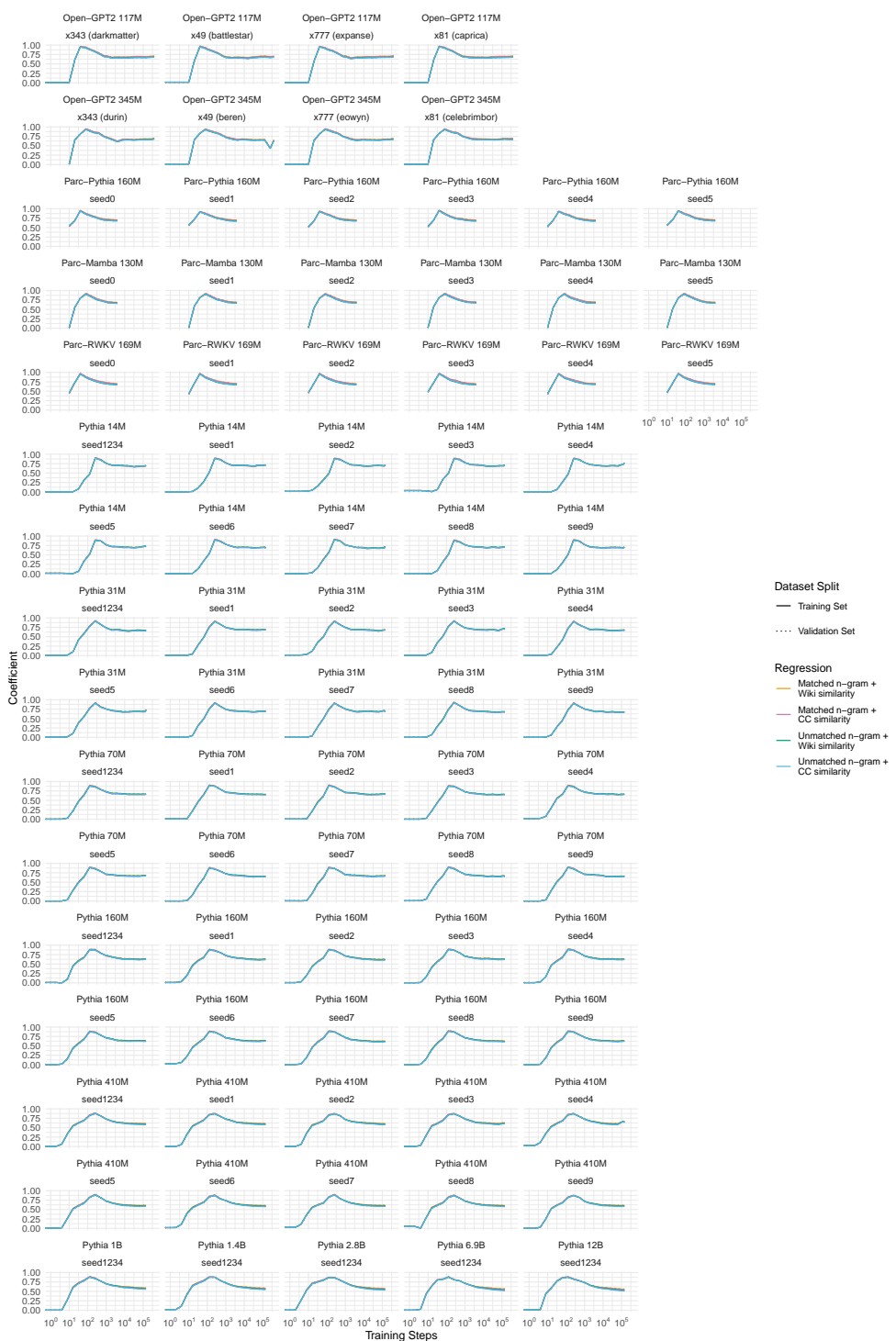

Figure 13: Proportion of the variance in language model log-probability explained by the regressions including uniformly-weighted similarity.

# K  Un-normalized Regression Coefficients

While the coefficients of the $z$-scored variables are easier to compare, they are less straightforward to interpret than those of un-normalized variables. For this reason, we carry out the same analysis with un-normalized variables. To allow the log-probabilities to be interpreted as bits, we convert them to the form $-\log_2(p)$. To ensure that all predictors share the same direction, we use cosine distance (i.e., $1 - \text{cosine similarity}$) rather than cosine similarity.

## K.1  SGPT-Weighted contextual semantic distance

We provide the coefficients of the $n$-gram (Figure 14) and contextual semantic distance (Figure 15) predictors for regressions with SGPT-weighted contextual semantic distance.

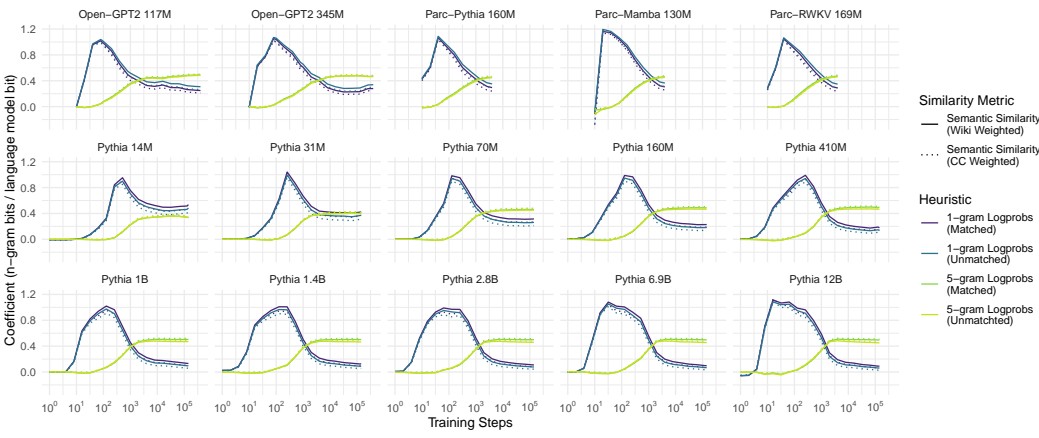

Figure 14: Un-normalized regression coefficients of unigram and 5-gram log-probability over the course of training under different conditions, specifically, whether the $n$-gram data is the same as that on which the language model was trained (matched) or not (unmatched), and whether SGPT-weighted contextual semantic similarity metric is calculated using Common-Crawl-based or Wikipedia-based fastText word vectors.

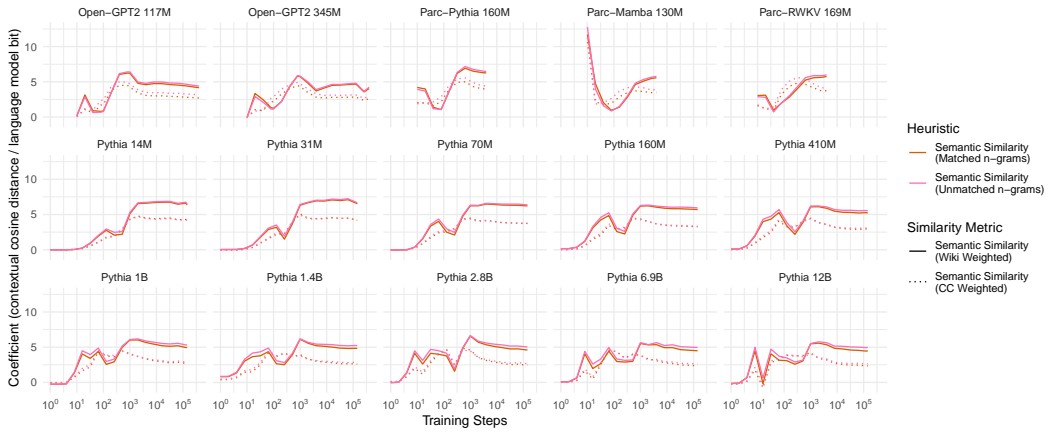

Figure 15: Un-normalized regression coefficients of contextual semantic similarity over the course of training under different conditions, specifically, whether the $n$-gram data is the same as that on which the language model was trained (matched) or not (unmatched), and whether SGPT-weighted contextual semantic similarity metric is calculated using Common-Crawl-based or Wikipedia-based fastText word vectors.

## K.2 Unweighted contextual semantic distance

We provide the coefficients of the $n$-gram (Figure 16) and unweighted contextual semantic distance (Figure 17) predictors for regressions with unweighted contextual semantic distance.

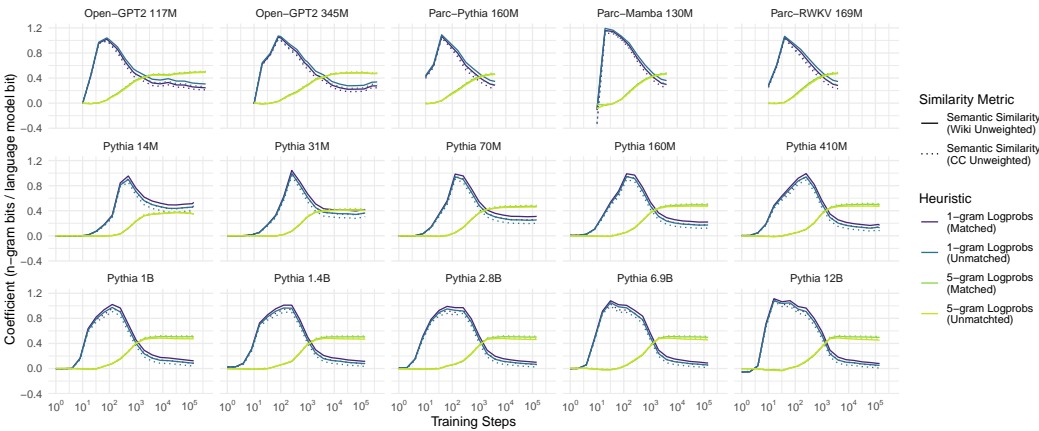

Figure 16: Un-normalized regression coefficients of unigram and 5-gram log-probability over the course of training under different conditions, specifically, whether the $n$-gram data is the same as that on which the language model was trained (matched) or not (unmatched), and whether unweighted contextual semantic similarity metric is calculated using Common-Crawl-based or Wikipedia-based fastText word vectors.

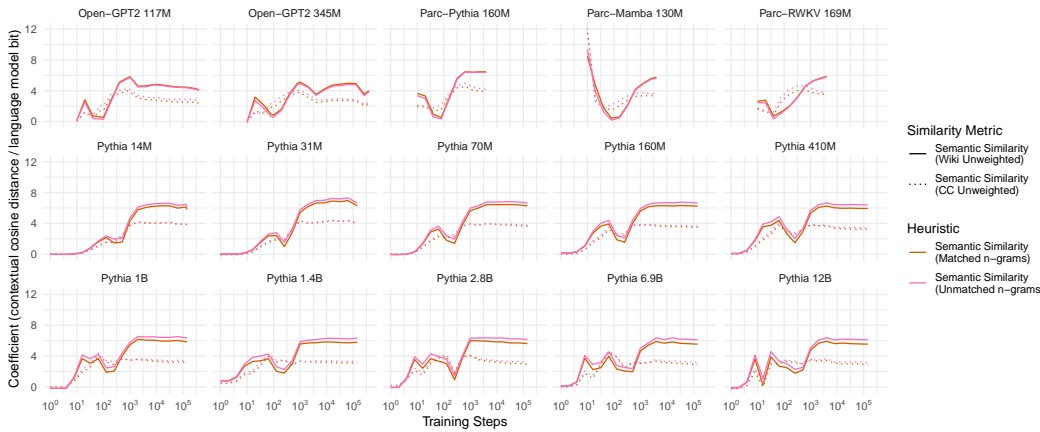

Figure 17: Un-normalized regression coefficients of contextual semantic similarity over the course of training under different conditions, specifically, whether the $n$-gram data is the same as that on which the language model was trained (matched) or not (unmatched), and whether unweighted contextual semantic similarity metric is calculated using Common-Crawl-based or Wikipedia-based fastText word vectors.

## L    Language Model Benchmark Performance

We evaluate the performance of all models used in the present study in two different ways. First, we calculate perplexity on the seven 'standard language modeling benchmarks' of Paloma (Magnusson et al., 2024), constructed from each of the following pre-existing pretraining datasets: C4 (Raffel et al., 2020), the English subset of mC4 (Chung et al., 2022), Dolma v1.5 (Soldaini et al., 2024),

RefinedWeb (Penedo et al., 2023), Penn Treebank (Marcus et al., 1999), RedPajama (Weber et al., 2024), and WikiText-103 (Merity et al., 2017). We plot the each model's perplexity on these text datasets in Figure 18 in bits-per-byte. All evaluations were carried out using the Language Model Evaluation Harness (Gao et al., 2021; Biderman et al., 2024).

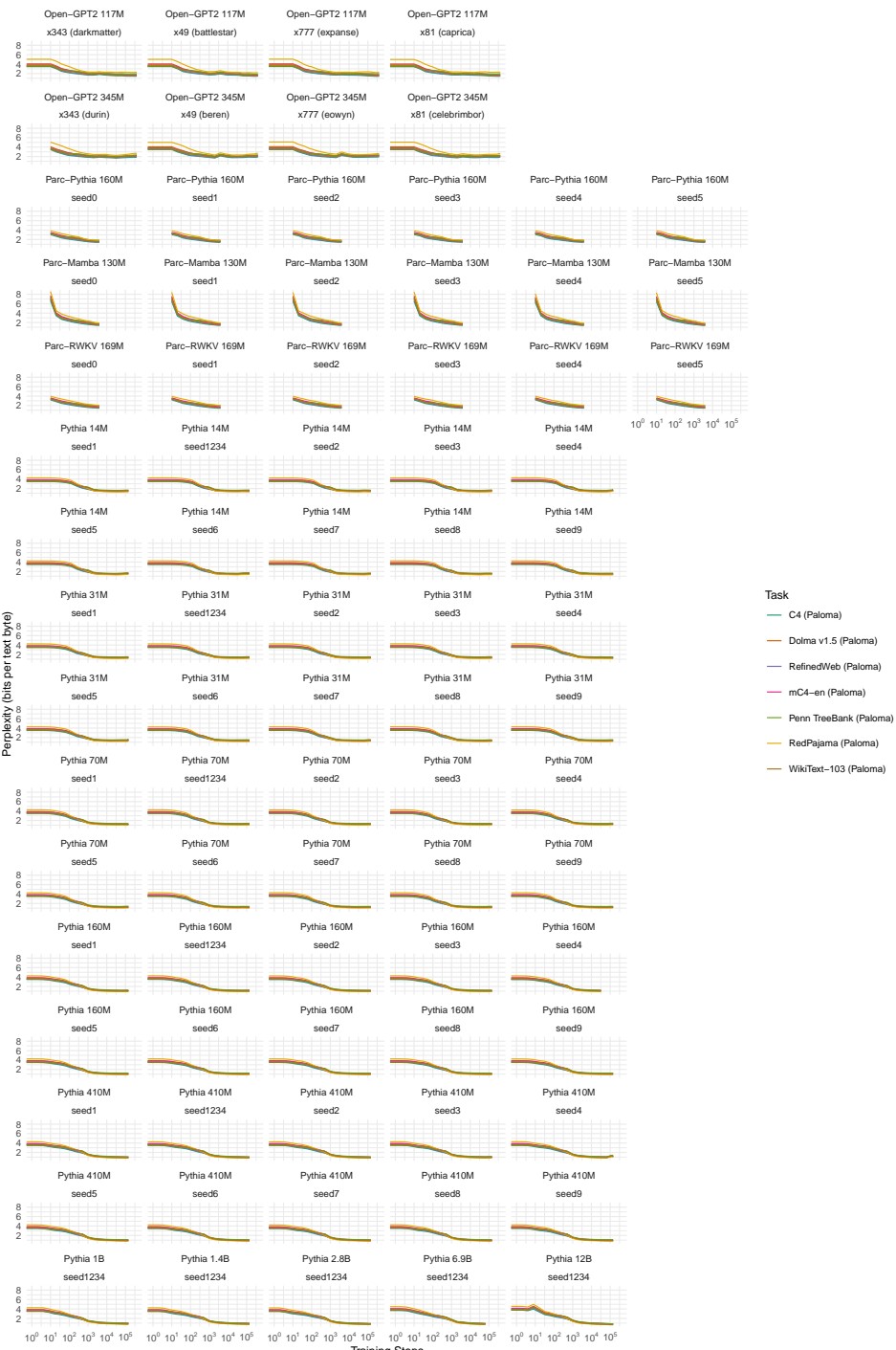

Figure 18: Seed-level perplexities of all models on 7 subsets of the Paloma dataset.

We also evaluate their performance of all models five benchmarks. To calibrate benchmark difficulty—if a benchmark is too difficult or too easy, it is not useful for comparing these models—we select

three benchmarks based on Biderman et al. (2023a), who find that clear differences based on model size and training data can be observed in the performance of the Pythia models on the OpenAI version of LAMBADA (Paperno et al., 2016; Radford et al., 2019), SCiQ (Welbl et al., 2017), and the Easy Set of the AI2 Reasoning Challenge (ARC; Clark et al., 2018). We additionally evaluate the models on SWAG (Zellers et al., 2018) and BLiMP (Warstadt et al., 2020). The accuracy and standard error of each model on each benchmark is provided in Figure 19.

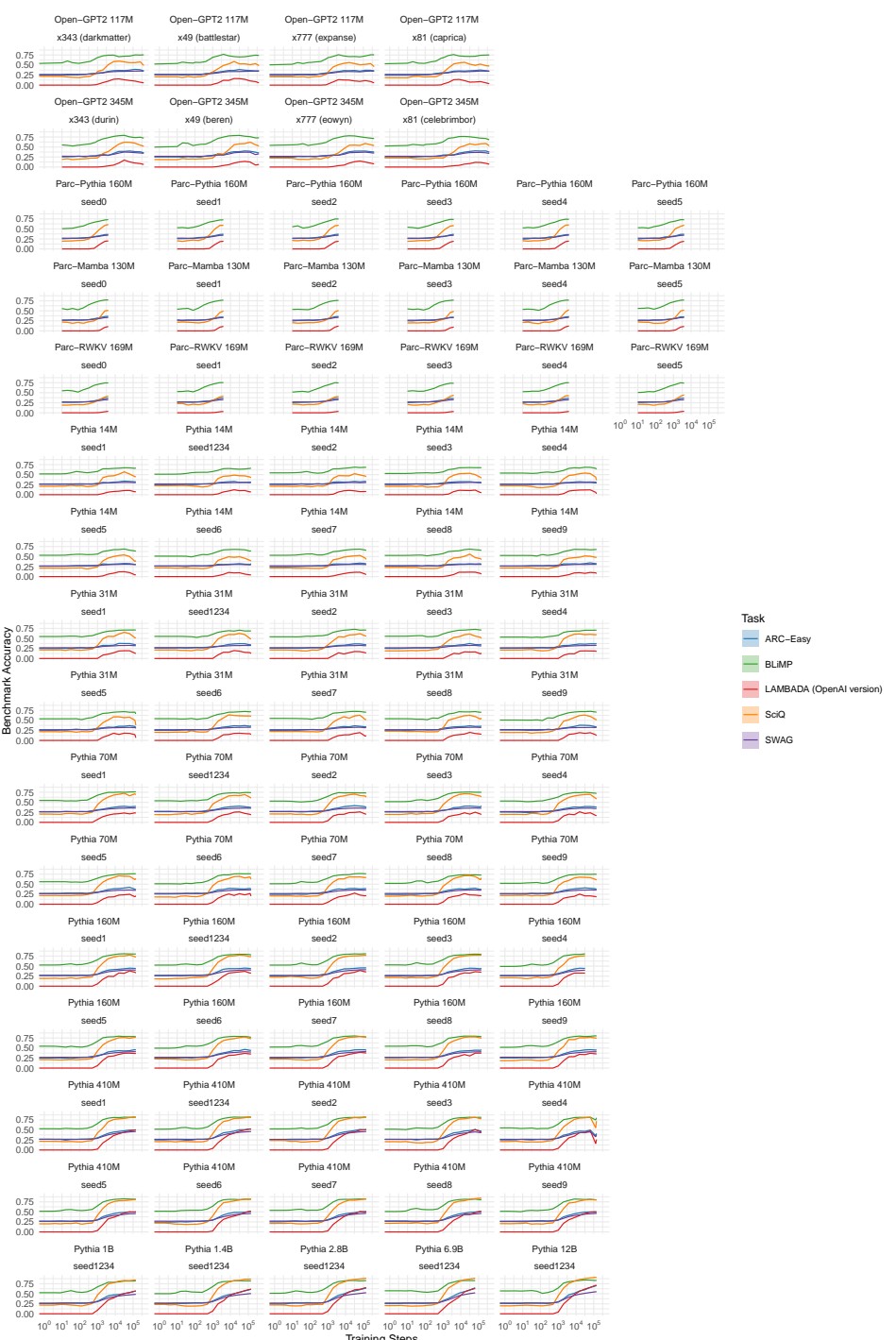

Figure 19: Seed-level accuracies of all models on 5 benchmarks, with shading used to indicate standard error.

