# OpenReview forum: "Language Model Behavioral Phases are Consistent Across Architecture, Training Data, and Scale"
_NeurIPS.cc/2025/Conference — NeurIPS 2025 poster_

### Official Review · Reviewer_Po9Y · 2025-07-02

**Clarity:** 3
**Significance:** 3
**Originality:** 2
**Rating:** 4
**Confidence:** 3

**Summary:**

The authors study the behavior of language models of 3 architectures and different scales as they are trained. They find that frequency of words, n-gram probabilities, and semantic similarly are highly indicative of the predicted probabilities.

**Questions:**

1) What is the training set used for exactly? Is this only for training the language models or just for the regression? Furthermore, how is it ensured that the sentences in L153-157 are not in any training dataset exactly?
2) The dataset mentioned focuses on words that are represented as a single token across models. Can we infer that this trend follows to more complex words?
3) I think a good addition to this work is to include implications of the findings. Would it be worth looking into better initialization on language models based on the insight of these training curves?
4) L332: On the one hand -> On one hand

**Ethical Concerns:**

["NO or VERY MINOR ethics concerns only"]

**Final Justification:**

The authors address my points, and I decide to raise my score to a borderline accept.

**Limitations:**

yes

**Paper Formatting Concerns:**

No formatting issues.

**Quality:**

3

**Strengths And Weaknesses:**

**Strengths**:
1) Well-motivated problem and interesting findings
2) Good related work and mentions of related work when necessary
3) Clear description of methodology and approach
4) Validated findings across a range of model sizes

**Weaknesses**:
1) I am unsure of the novelty of this work. Is the contribution the systematic training and validation of the correlation between these heuristics, suggested in previous work, and the assigned word probabilities in language models? The value of the introduced dataset is unclear to me.
2) Please refer to questions below.

I am looking forward to hearing back from the authors and adjusting my scores accordingly.

---

> ### Author Rebuttal · Authors · 2025-07-30
>
> We thank the reviewer for their comments and questions, which we address below:
>
> > I am unsure of the novelty of this work. Is the contribution the systematic training and validation of the correlation between these heuristics, suggested in previous work, and the assigned word probabilities in language models?
>
> We believe that the main novel contributions of this study are the following findings:
>
> 1. The extent to which models do most closely correlate with n-grams of increasing order over the course of training decreases with each order.
>
> 2. While language model predictions correlate with n-grams of increasing order over the course of training, their predictions are still biased towards lower-order n-grams relative to these (i.e., unigrams explain language model behavior above and beyond 5-grams).
>
> 3. Except in the very early stages of training, language model predictions correlate with semantic similarity, and the effect of this is relatively consistent throughout training.
>
> 4. The variance in language model log-probability explained by three simple heuristics (unigram log-probability, 5-gram log-probability, and word-level semantic similarity) is substantial for all models after the first 10-100 steps (depending on the model), and explains over half (and up to 98%) of their behavior, with the extent to which this is the case over the course of training showing consistent patterns.
>
> 5. The above patterns are consistent across size and architecture, suggesting that they may reflect more general or even universal properties of systems that are trained to predict language.
>
> > The value of the introduced dataset is unclear to me.
>
> The main purpose of our dataset was to provide a set of words in context that are in natural sentences of English and that are not included in the training data of any of the models analyzed.
>
> > What is the training set used for exactly? Is this only for training the language models or just for the regression? Furthermore, how is it ensured that the sentences in L153-157 are not in any training dataset exactly?
>
> The training set of the introduced dataset was used to fit the regression only, thus allowing us to compare fit on the validation set to ensure that the regression didn’t overfit. We ensured that the sentences in the training set were not included in the model training sets using the infini-gram tool, which allows users to test whether a given text sequence of any length occurs in a dataset. We locally implemented infini-gram for the OpenWebText corpus, and used the built-in API for The Pile corpus. The details of this are provided in Appendix D; however, we agree that this is important and will include them in the main text of the revised paper.
>
> > The dataset mentioned focuses on words that are represented as a single token across models. Can we infer that this trend follows to more complex words?
>
> This is an interesting question. We constrained our analysis to such words because our language models of interest, n-gram models, and the fasttext models we used all had different tokenizers. We do not have any direct evidence from our own work, but one possibly relevant study is that of Chang et al. (2024), who look at n-gram and transformer language models with the same tokenizers and include tokens that occur as part of larger words. Like us, they find that transformer log-probabilities correlate more with n-gram log-probabilities of increasing order over the course of training, suggesting that at least this pattern is likely to be consistent, but they do find more variability in the log-probabilities assigned to tokens in multi-token words (especially word-intermediate and word-final tokens) across different runs of training. We suspect that the semantic similarity results would similarly be relatively robust but with more variability for tokens in multi-token words.
>
> > I think a good addition to this work is to include implications of the findings. Would it be worth looking into better initialization on language models based on the insight of these training curves?
>
> As we note in our response to Reviewer 93xh, in combination with previous work the training dynamics of traditional RNNs (Karpathy et al., 2016) and the downstream performance of Pythia models over the course of training (Biderman et al., 2023), our results may suggest that the patterns we observe are prerequisite stages in learning that must occur before the more complex patterns that underlie good performance at downstream tasks can be learned; and thus, that speeding up the learning of n-grams and semantic similarity (or possibly initializing the models such that their predictions reflect these) could in principle lead to more efficient training.
>
> > L332: On the one hand -> On one hand
>
> We thank the reviewer for noting this typo.
>
>
> **References:**
>
> - Biderman, S., Schoelkopf, H., Anthony, Q. G., Bradley, H., O’Brien, K., Hallahan, E., Khan, M. A., Purohit, S., Prashanth, U. S., Raff, E., Skowron, A., Sutawika, L., & Wal, O. V. D. (2023). Pythia: A Suite for Analyzing Large Language Models Across Training and Scaling. *Proceedings of the 40th International Conference on Machine Learning*, 2397–2430. https://proceedings.mlr.press/v202/biderman23a.html
> - Chang, T. A., Tu, Z., & Bergen, B. K. (2024). Characterizing Learning Curves During Language Model Pre-Training: Learning, Forgetting, and Stability. *Transactions of the Association for Computational Linguistics, 12*, 1346–1362. https://doi.org/10.1162/tacl_a_00708
> - Karpathy, A., Johnson, J., & Fei-Fei, L. (2016, February 18). Visualizing and Understanding Recurrent Networks. *International Conference on Learning Representations (Workshop Track)*. https://openreview.net/forum?id=71BmK0m6qfAE8VvKUQWB

---

> > ### Comment · Reviewer_Po9Y · 2025-08-05
> >
> > The authors address my concerns, I have no other questions.

---

### Official Review · Reviewer_93xh · 2025-07-02

**Clarity:** 4
**Significance:** 3
**Originality:** 2
**Rating:** 5
**Confidence:** 3

**Summary:**

This paper investigates if language model behavior can be explained by simple features like n-gram probabilities and embedding-similarity between tokens and their context. While similar studies have been done in the past, this one expands the scope by studying transformer models of a range of sizes, as well as models of other architectures (Mamba and RWKV). Particularly, the analyses performed in this work are 1) measuring the correlation between LM's log-probabilities and each of the simpler features; and 2) measuring the fit of a regression model that predicts the logprobs from those features, along with the significance of the coefficients. The findings from this work are aligned with prior work.

**Questions:**

Can you please list some potential actionable insights for LM developers based on the findings in this work?

**Ethical Concerns:**

["NO or VERY MINOR ethics concerns only"]

**Final Justification:**

I find the authors' response informative. The discussion has not changed my assessment of the paper.

**Limitations:**

Yes

**Paper Formatting Concerns:**

Citation format in the paper is LastName et al.

**Quality:**

3

**Strengths And Weaknesses:**

### Strengths

The scale of the study is larger than prior work, and the results show that the findings generalize across architectures.

### Weaknesses

The study is very similar to prior work (as is made clear in the paper). While it is nice to see that the findings are aligned with prior work, it is not clear what new actionable insights can be drawn from this work. A discussion on how LM developers can use these findings to improve the LM training procedure would be helpful.

---

> ### Author Rebuttal · Authors · 2025-07-30
>
> We would like to thank the reviewer for their feedback. We address the weaknesses and questions below:
>
> > The study is very similar to prior work (as is made clear in the paper). While it is nice to see that the findings are aligned with prior work, it is not clear what new actionable insights can be drawn from this work.
>
> Our study highlights that except for in the very earliest stages of training, language model predictions tend to favor tokens with high n-gram probability and word-level semantic similarity. Previous work has demonstrated correlations with these variables, but we show that, at least after the initial 100 or so steps of training, these heuristics can explain 50-98% of the variance in the log-probabilities calculated by language models. This directly relates to previous work that has (implicitly) shown cases where a tendency to make predictions in line with these heuristics leads to undesirable behavior. For example, researchers have found that language models will often complete famous quotes or sayings even when prompted not to (McKenzie et al., 2023), which suggests a strong bias towards frequent n-gram continuations. Similarly, (Gonen et al., 2025) show that language models display ‘semantic leakage’, where they will often generate text that is inappropriately semantically related to words in the input, e.g., “His name is **Cedar**. His friend lives in **a treehouse**” (Llama2-70b-chat completion bolded).
>
> Thus, our work provides a more precise and generalizable characterization of the patterns underlying these and related issues with language model behavior, as well as a method for characterizing them in a given model. In terms of actionable insights, then, the analysis methods we demonstrate in the present study may make it possible to predict which models are more susceptible to such biases in their generation behavior before deployment and without relying on more specific tests that are unlikely to cover all possible phenomena. We will include a discussion of this in the revised paper.
>
>
> > A discussion on how LM developers can use these findings to improve the LM training procedure would be helpful.
>
> > Can you please list some potential actionable insights for LM developers based on the findings in this work?
>
> In combination with the results of research on traditional RNNs, which also become sensitive to the statistics of increasingly long contexts over the course of training (Karpathy et al., 2016), our results suggest that the patterns we observe may reflect more general (and possibly even universal) stages of learning for sufficiently powerful language models of any size or architecture. When we compare our results with work looking at the performance of the Pythia models at downstream tasks at various points over the course of training (Biderman et al., 2023), we see that the models begin to improve at these tasks only after the point at which we see correlations between language model log-probabilities and our heuristics begin to plateau or decrease. This could suggest that learning (at least implicitly) higher-order n-grams and semantic similarity may be a prerequisite for learning the more complex patterns that underlie good performance at downstream tasks; and thus, that speeding up the learning of these heuristics (or even initializing the models such that they align with these heuristics) could lead to more efficient training. This is something we plan to discuss in our revised paper, and plan to further develop by evaluating all the models included in our study on a wider range of benchmarks.
>
>
> **References:**
> - Biderman, S., Schoelkopf, H., Anthony, Q. G., Bradley, H., O’Brien, K., Hallahan, E., Khan, M. A., Purohit, S., Prashanth, U. S., Raff, E., Skowron, A., Sutawika, L., & Wal, O. V. D. (2023). Pythia: A Suite for Analyzing Large Language Models Across Training and Scaling. *Proceedings of the 40th International Conference on Machine Learning*, 2397–2430. https://proceedings.mlr.press/v202/biderman23a.html
> - Gonen, H., Blevins, T., Liu, A., Zettlemoyer, L., & Smith, N. A. (2025). Does Liking Yellow Imply Driving a School Bus? Semantic Leakage in Language Models. In L. Chiruzzo, A. Ritter, & L. Wang (Eds.), *Proceedings of the 2025 Conference of the Nations of the Americas Chapter of the Association for Computational Linguistics: Human Language Technologies (Volume 1: Long Papers)* (pp. 785–798). Association for Computational Linguistics. https://doi.org/10.18653/v1/2025.naacl-long.35
> - Karpathy, A., Johnson, J., & Fei-Fei, L. (2016, February 18). Visualizing and Understanding Recurrent Networks. *International Conference on Learning Representations (Workshop Track)*. https://openreview.net/forum?id=71BmK0m6qfAE8VvKUQWB
> - McKenzie, I. R., Lyzhov, A., Pieler, M. M., Parrish, A., Mueller, A., Prabhu, A., McLean, E., Shen, X., Cavanagh, J., Gritsevskiy, A. G., Kauffman, D., Kirtland, A. T., Zhou, Z., Zhang, Y., Huang, S., Wurgaft, D., Weiss, M., Ross, A., Recchia, G., … Perez, E. (2023). Inverse Scaling: When Bigger Isn’t Better. *Transactions on Machine Learning Research*. https://openreview.net/forum?id=DwgRm72GQF

---

### Official Review · Reviewer_xNHd · 2025-07-03

**Clarity:** 3
**Significance:** 3
**Originality:** 3
**Rating:** 4
**Confidence:** 5

**Summary:**

This article analyzes the training dynamics of different architectures in the early stages of training and finds that different architectures will first learn word frequency, n-gram syntax, and context similarity.

**Questions:**

1. N-grams are a fairly low-level feature, and the intelligence shown by recent large models, such as contextual learning capabilities such as induction heads, is often not in n-grams. I'm not sure how much this helps reveal about the intelligence of the model.  There are some articles about the learning dynamics of n-gram syntax and induction heads for transformers [1][2], which I feel are very relevant to what this paper is about. I think the paper can discuss them.

2. For transformer, mamba and rwkv, their performance in long contexts shows significant differences. However, the indicators measured in this paper seem to be irrelevant to long contexts, so they show greater similarity. What do you think about this?

3. I think the cross-architecture comparison is a highlight of the article, and I also think comparing more powerful architectures would increase the scope of the article. From the perspective of theoretical complexity, transformer, mamba, and rwkv-4 are limited by the expressiveness of TC^0. However, in the past year, some models have broken through the expressiveness limit of TC^0, such as DeltaNet [1], RWKV-7[2], Path[3], and Deltaformer[4], which can perform state tracking tasks such as S5. I am not sure whether such architectures also have learning dynamics similar to those mentioned in this article.

4. I think this article can also discuss broader impacts, such as potential insight for llm training. Starting from this article, do we need to improve the model structure or optimization methods to enable the model to better learn n-grams and quickly move on to the next stage of learning? Or, if an inductive bias for learning n-grams is introduced, it will make the model more inclined to learn n-grams and difficult to move on to the next stage of learning?




[1] Birth of a transformer: A memory viewpoint. NeurIPS.

[2] How Transformers Get Rich: Approximation and Dynamics Analysis. Arxiv 2024.

[3] Linear transformers are secretly fast weight programmers. ICML 2021.

[4] RWKV-7 "Goose" with Expressive Dynamic State Evolution. Arxiv 2025.

[5] PaTH Attention: Position Encoding via Accumulating Householder Transformations. Arxiv 2025.

[6] Understanding Transformer from the Perspective of Associative Memory. Arxiv 2025.

**Ethical Concerns:**

["NO or VERY MINOR ethics concerns only"]

**Final Justification:**

I think this is a valuable empirical article, but I am not sure if it is sufficient for the NeurIPS. So I tend to weak accept.

**Limitations:**

See weakness.

**Quality:**

3

**Strengths And Weaknesses:**

**Strengths**
1. The experimental results are detailed. The paper compares the learning dynamics of different architectures and finds many similarities. The cross-architecture comparison fills the gap.

2. The paper is clear and easy to follow.

3. The references are rich, with detailed discussions of related work.

**Weakness**

1. Some details are not clear in the paper.  For example, rwkv and mamba, as core architectures, have had more than one generation of versions, and there are differences between them. The text does not specify which version they are. We have to find out which version is used from the supplementary materials.

2. Novelty is limited. Rather than discovering a novel idea, this paper is more about verifying that an existing idea still holds true in a different architecture.

---

> ### Author Rebuttal · Authors · 2025-07-30
>
> We would like to thank the reviewer for their detailed questions and feedback, which we address below:
>
> > Some details are not clear in the paper. For example, rwkv and mamba, as core architectures, have had more than one generation of versions, and there are differences between them. The text does not specify which version they are. We have to find out which version is used from the supplementary materials.
>
> We thank the reviewer for noting this. We will clarify the versions (RWKV-4 and Mamba 1) in the revised paper.
>
> > Novelty is limited. Rather than discovering a novel idea, this paper is more about verifying that an existing idea still holds true in a different architecture.
>
> We agree that several of the motivating ideas (that language models learn n-grams and predict words that are semantically similar to be more likely than dissimilar words) have been observed and discussed in previous work, but we do believe that our study makes several novel contributions  to our understanding of the training dynamics of language models in addition to showing that the results are consistent across architectures:
>
> 1. We show that while models do most closely correlate with n-grams of increasing order over the course of training, the extent to which this is the case decreases with each order.
>
> 2. We provide evidence that while language model predictions correlate with n-grams of increasing order over the course of training, their predictions are still biased towards lower-order n-grams relative to these (i.e., unigrams explain language model behavior above and beyond 5-grams). This may partially explain (1), but even if it does not fully explain the pattern, we believe that together these findings provide a valuable starting point for investigating which other heuristics language model predictions may correlate with in the later stages of training, perhaps in combination with mechanistic interpretability analyses of the heuristics we investigate.
>
> 3. We provide evidence that except in the very early stages of training, language model predictions correlate with semantic similarity, and the effect of this is relatively consistent throughout training.
>
> 4. We provide an estimate of how much variance in language model probability is explained by these simple heuristics, show that it is substantial for all models after the first 10-100 steps (consistently over 50% and up to 98%), and that there is a predictable trend to the amount of variance explained.
>
> Additionally, we believe that the fact that these patterns are seen despite the differences in architecture is theoretically important in itself, as it may suggest that they reflect more general (or possibly even universal) properties of systems that are trained to predict language incrementally.
>
> > N-grams are a fairly low-level feature, and the intelligence shown by recent large models, such as contextual learning capabilities such as induction heads, is often not in n-grams. I'm not sure how much this helps reveal about the intelligence of the model. There are some articles about the learning dynamics of n-gram syntax and induction heads for transformers [1][2], which I feel are very relevant to what this paper is about. I think the paper can discuss them.
>
> We thank the reviewer for bringing these papers to our attention. We agree that they are relevant to the current work, and will include a discussion of them (particularly, what is already known about the relationship between global and in-context n-gram prediction and the mechanisms underlying them in transformers) in the revised paper.
>
> > For transformer, mamba and rwkv, their performance in long contexts shows significant differences. However, the indicators measured in this paper seem to be irrelevant to long contexts, so they show greater similarity. What do you think about this?
>
> This is an interesting point. While there are clear differences between the architectures, it is also true that their behavior is often comparable on downstream tasks (even if there are some differences), as seen in the evaluations reported in the original Mamba and RWKV papers (Gu & Dao, 2024; Peng et al., 2023). Thus, despite their very different architectures, their behavior appears to converge to similar patterns much of the time, which is an interesting implication of our findings (along with previous work) in its own right. With respect to long contexts in particular, the extent to which in practice the capability to deal with very long contexts is important is an open question and likely to vary by domain—for example, it has been argued that there is a general pressure in natural language to have shorter-range dependencies (see, e.g., Futrell et al., 2020); thus, differences in performance on longer-range dependencies may not have as much of an impact on behavior in this domain in practice. In any case, our results help to describe the patterns that occur over shorter contexts (and that appear to be common across models of different architectures), and thus provides a starting point for analyses of longer contexts, other more complex phenomena, and comparisons across architectures.
>
> > I think the cross-architecture comparison is a highlight of the article, and I also think comparing more powerful architectures would increase the scope of the article. From the perspective of theoretical complexity, transformer, mamba, and rwkv-4 are limited by the expressiveness of TC^0. However, in the past year, some models have broken through the expressiveness limit of TC^0, such as DeltaNet [1], RWKV-7[2], Path[3], and Deltaformer[4], which can perform state tracking tasks such as S5. I am not sure whether such architectures also have learning dynamics similar to those mentioned in this article.
>
> We thank the reviewer for bringing this exciting development to our attention. We agree that a comparison of training dynamics across architectures with different levels of expressiveness would likely be interesting and informative. However, much of the motivation for using such models, as well as the empirical improvements observed, appear to be for long-context settings, and since our analysis focuses on single sentence contexts, we would not expect a substantial difference in the trends observed. Nonetheless, we remain excited in the possibility of such comparisons in future work.
>
> > I think this article can also discuss broader impacts, such as potential insight for llm training. Starting from this article, do we need to improve the model structure or optimization methods to enable the model to better learn n-grams and quickly move on to the next stage of learning? Or, if an inductive bias for learning n-grams is introduced, it will make the model more inclined to learn n-grams and difficult to move on to the next stage of learning?
>
> We agree that this is an interesting question. If we consider our results for the Pythia models in combination with their downstream performance on tasks as reported in the original Pythia paper (Biderman et al., 2023), it appears that the point at which language model correlation with n-grams of order n≥3 begins to plateau (steps 1,000-2,000; which for the Pythia models corresponds to 2B-4B tokens) is also the point at which they begin to start improving substantially at several downstream tasks: LAMBADA, SciQ, and the Easy set of the AI2 Reasoning Challenge; with the models that end up showing the lowest correlation with n-grams performing the best. Thus, the results of our study could be taken to suggest that learning n-grams may be a prerequisite (but not sufficient) for downstream task performance, and thus, that a training procedure that allows them to be learned more quickly and then move on to the next stage would be the most optimal approach. How exactly this could be implemented is a question for future work. In addition to including a discussion of this in the revision of our paper, we also plan to evaluate all the models included in our study on a wider range of downstream tasks, and thus hopefully gain a better understanding of how the extent to which model performance correlates with the heuristics we look at correlates with downstream performance. In preliminary testing of our models (OWT-Pythia, OWT-Mamba, and OWT-RWKV) on the LAMBADA benchmark, we find that they show the same pattern—their correlation with n-grams of order n≥3 begins to plateau by step 2650, and this is also the point at which we see the improvement at the benchmark begin.
>
> **References:**
>
> - Biderman, S., Schoelkopf, H., Anthony, Q. G., Bradley, H., O’Brien, K., Hallahan, E., Khan, M. A., Purohit, S., Prashanth, U. S., Raff, E., Skowron, A., Sutawika, L., & Wal, O. V. D. (2023). Pythia: A Suite for Analyzing Large Language Models Across Training and Scaling. *Proceedings of the 40th International Conference on Machine Learning*, 2397–2430. https://proceedings.mlr.press/v202/biderman23a.html
> - Futrell, R., Levy, R. P., & Gibson, E. (2020). Dependency locality as an explanatory principle for word order. *Language, 96*(2), 371–412. https://doi.org/10.1353/lan.2020.0024
> - Gu, A., & Dao, T. (2024, August 26). Mamba: Linear-Time Sequence Modeling with Selective State Spaces. First Conference on Language Modeling. https://openreview.net/forum?id=tEYskw1VY2
> - Peng, B., Alcaide, E., Anthony, Q., Albalak, A., Arcadinho, S., Biderman, S., Cao, H., Cheng, X., Chung, M., Derczynski, L., Du, X., Grella, M., Gv, K., He, X., Hou, H., Kazienko, P., Kocon, J., Kong, J., Koptyra, B., … Zhu, R.-J. (2023). RWKV: Reinventing RNNs for the Transformer Era. In H. Bouamor, J. Pino, & K. Bali (Eds.), *Findings of the Association for Computational Linguistics: EMNLP 2023* (pp. 14048–14077). Association for Computational Linguistics. https://doi.org/10.18653/v1/2023.findings-emnlp.936

---

> > ### Comment · Reviewer_xNHd · 2025-08-04
> >
> > Thank you very much for the detail response, I hope adding relevant content from the rebuttal response in the revision can enhance the paper. I believe this is a valuable empirical paper and I will maintain my score but increase my confidence.

---

### Official Review · Reviewer_8SRQ · 2025-07-05

**Clarity:** 4
**Significance:** 3
**Originality:** 4
**Rating:** 5
**Confidence:** 3

**Summary:**

This work aims to investigate the inner mechanisms of LLMs in feature extraction during training from the perspective of interpretability.
Based on the analysis on extensive model checkpoints and datasets, this paper highlights that, no matter the model type or training setup, the behavior of these models at the word level can be largely explained by just three heuristics: word-frequency, n-gram probability, and semantic similarity.
They found that these three factors together could explain up to 98% of the variation in the probability that a model assigns to a word in a sentence. Specifically, they observed that as models train, their behavior shifts  in predictable "phases," starting by overfitting to simple patterns (like single-word frequency), then learning more complex n-gram patterns, and becoming sensitive to meaning/semantics early in training.
Moreover, the work reveals that there exists remarkable and consistent patterns across different architectures (like transformers, RWKV, and Mamba), datasets, and model sizes.
This can facilitate our understanding of how LLMs capture statistical features during training and leverage them in prediction.

**Questions:**

1. This work shows that training patterns are largely consistent across different model architectures. However, in practical applications, LLMs from different families often exhibit varying levels of performance on downstream tasks. How can you explain this discrepancy? Which capability of LLMs can be affected by these patterns?
2. Figure 3 shows that Pythia models with larger model sizes are less explainable using these three heuristics. Can you further figure out what other significant mechanisms/heuristics (e.g., syntactic features) are at play in larger-sized LLMs? Such mechanisms/heuristics may better explain the performance discrepancies between LLMs across various downstream tasks.

**Ethical Concerns:**

["NO or VERY MINOR ethics concerns only"]

**Final Justification:**

The authors have addressed my concerns.

**Limitations:**

Yes

**Quality:**

3

**Strengths And Weaknesses:**

Strengths:
1. This work investigates the learning behaviors of LLMs from three heuristics, word-frequency, n-gram probability, and semantic similarity, the combination of which can better explain the prediction of LLMs.
2. As word-frequency and n-gram probabilities based on counts do not take word or phrase similarity in consideration, the introduction of semantic similarity on top of them can better enhance the explainability of LLM prediction behaviors.

Weaknesses:
1. The analysis focuses on n-grams up to size 5, which may limit the finding of more useful patterns. Language may involve longer-range dependencies, especially in larger, more powerful models.
2. This work may basically explain how LLMs leverage ngrams for prediction in early training stages. However, as Figure 3 shows, even with these three heuristics, there’s still unexplained variance in model behavior, especially for the largest models and in later training stages, suggesting that more complex or subtle mechanisms are at play.

---

> ### Author Rebuttal · Authors · 2025-07-30
>
> We thank the reviewer for their feedback, and address the weaknesses and questions below:
>
> > The analysis focuses on n-grams up to size 5, which may limit the finding of more useful patterns. Language may involve longer-range dependencies, especially in larger, more powerful models.
>
> > This work may basically explain how LLMs leverage ngrams for prediction in early training stages. However, as Figure 3 shows, even with these three heuristics, there’s still unexplained variance in model behavior, especially for the largest models and in later training stages, suggesting that more complex or subtle mechanisms are at play.
>
> > Figure 3 shows that Pythia models with larger model sizes are less explainable using these three heuristics. Can you further figure out what other significant mechanisms/heuristics (e.g., syntactic features) are at play in larger-sized LLMs? Such mechanisms/heuristics may better explain the performance discrepancies between LLMs across various downstream tasks.
>
> We acknowledge the reviewer’s concern that there can be dependencies in language longer than those that can be captured by 5-grams, and that our approach leaves some of the variance in model behavior unexplained for larger models later in training. However, our goal in this paper is not to explain all of language model behavior with simple heuristics—if this were possible, there would be no need for contemporary language models in the first place. Instead, our goal in this study is to identify general patterns in the behavior of language models over the course of training across scale and architecture and to see how universal they are.
>
> Our results demonstrate that there is in fact a consistent pattern in the extent to which language model probabilities correlate with the simple heuristics of n-gram probability and word-level semantic similarity, and that after the very early stages of training, the three heuristics we include in our regression (unigram probability, 5-gram probability, and word-level semantic similarity) can explain over 50% of the variance in language model behavior, and at some stages, over 90%. Our hope is that our results can serve as a starting point or a baseline for investigating the more complex patterns of language model behavior that go beyond this.
>
> > This work shows that training patterns are largely consistent across different model architectures. However, in practical applications, LLMs from different families often exhibit varying levels of performance on downstream tasks. How can you explain this discrepancy? Which capability of LLMs can be affected by these patterns?
>
> We thank the reviewer for this question, and would be interested if the reviewer could provide more details about which specific findings they are referring to. As far as we are aware, in many cases, the differences in performance that can be attributed to architecture alone (rather than, e.g., number of parameters or training data) are not as large as might be expected. For example, while we do see a difference between the performance of Mamba, RWKV, and Pythia models in the original RWKV (Peng et al., 2023) and Mamba (Gu & Dao, 2024) papers, it is not drastic. Additionally, as previously noted, one of our goals in this work is to find commonalities across architectures, which provides a starting point from which to analyze differences.
>
> Inspired by this reviewer question, we have carried out further analyses to look at similarities across the models that we trained (i.e., the Mamba, RWKV, and Pythia models trained on OpenWebText) over the course of training. We find that from checkpoint 80 onward, all the log-probabilities of all models are highly correlated—all pairwise comparisons of the Pearson correlation between the log-probabilities calculated by the models have r≥0.93. This again suggests, as in the previous work we note, that in many cases, behavior across architectures is comparable. Returning to the question of differences between architectures observed in previous work, we are not sure which findings precisely the reviewer is referring to, but it is worth noting that studies investigating such questions are likely to explicitly target adversarial and edge cases, and thus, situations in which the differences are larger may simply be rare, and therefore not account for the majority of language model behavior.
>
> **References:**
> - Gu, A., & Dao, T. (2024). Mamba: Linear-Time Sequence Modeling with Selective State Spaces. *First Conference on Language Modeling*. https://openreview.net/forum?id=tEYskw1VY2
> - Peng, B., Alcaide, E., Anthony, Q., Albalak, A., Arcadinho, S., Biderman, S., Cao, H., Cheng, X., Chung, M., Derczynski, L., Du, X., Grella, M., Gv, K., He, X., Hou, H., Kazienko, P., Kocon, J., Kong, J., Koptyra, B., … Zhu, R.-J. (2023). *RWKV: Reinventing RNNs for the Transformer Era. In H. Bouamor, J. Pino, & K. Bali (Eds.), *Findings of the Association for Computational Linguistics: EMNLP 2023* (pp. 14048–14077). Association for Computational Linguistics. https://doi.org/10.18653/v1/2023.findings-emnlp.936

---

> > ### Comment · Reviewer_8SRQ · 2025-08-09
> >
> > Thank you very much for your responses regarding my questions and concerns. They are explicit.
> > You have demonstrated that those heuristics can explain model log-probabilities, namely,  next-token prediction,  to a large extent.
> > Regarding the discrepancy in downstream task performance, I am curious about whether your findings regarding the heuristics can be further employed to explain their (potential) performance gaps in language modeling, reasoning, etc.

---

### Decision · Program_Chairs · 2025-09-17

**Decision:**

Accept (poster)

**Comment:**

This paper investigates interpretability in the training dynamics of LLM across multiple architectures. The central claim is that LLM prediction behavior at the word level can largely be explained by three simple heuristics: word frequency, n-gram probability, and semantic similarity. The authors show that these heuristics together explain up to 98% of the variance in model predictions during training. They also identify predictable training “phases”: models begin with frequency-based prediction, progress to n-gram reliance, and then incorporate semantic similarity. Importantly, these patterns are found to be consistent across architectures and scales, suggesting a potentially universal property of LLM training.

Overall, the paper makes a solid empirical contribution by showing that simple heuristics explain a substantial portion of LLM behavior across architectures. These findings provide a baseline framework for future interpretability studies and suggest possible implications for more efficient training.